# Dermatophytosis in Companion Animals in Portugal: A Comprehensive Epidemiological Retrospective Study of 12 Years (2012–2023)

**DOI:** 10.3390/microorganisms12081727

**Published:** 2024-08-22

**Authors:** Ricardo Lopes, Andreia Garcês, Augusto Silva, Paula Brilhante-Simões, Ângela Martins, Luís Cardoso, Elsa Leclerc Duarte, Ana Cláudia Coelho

**Affiliations:** 1Department of Veterinary Sciences, University of Trás–os–Montes e Alto Douro (UTAD), 5000–801 Vila Real, Portugal; lopes.rmv@gmail.com (R.L.); lcardoso@utad.pt (L.C.); 2Department of Veterinary and Animal Sciences, University Institute of Health Sciences (IUCS), CESPU, 4585–116 Gandra, Portugal; paulabrilhante@inno.pt; 3Wildlife Rehabilitation Centre (CRAS), Veterinary Teaching Hospital, University of Trás–os–Montes e Alto Douro (UTAD), 5000–801 Vila Real, Portugal; andreiamvg@gmail.com; 4Animal and Veterinary Research Centre (CECAV), Associate Laboratory for Animal and Veterinary Sciences (AL4AnimalS), University of Trás–os–Montes e Alto Douro (UTAD), 5000–801 Vila Real, Portugal; angela@utad.pt; 5INNO Veterinary Laboratories, R. Cândido de Sousa 15, 4710–300 Braga, Portugal; augustosilva@inno.pt; 6Department of Veterinary Medicine, School of Science and Technology, University of Évora, Polo da Mitra, Apartado 94, 7002–554 Évora, Portugal; emld@uevora.pt; 7Mediterranean Institute for Agriculture, Environment and Development (MED), Global Change and Sustainability Institute (CHANGE), University of Évora, Polo da Mitra, Apartado 94, 7002–554 Évora, Portugal

**Keywords:** clinical pathology, companion animals, dermatophyte, epidemiology, fungi, One Health, Planetary Health, skin infection, Stockholm Paradigm, zoonosis

## Abstract

Dermatophytosis, commonly referred to as ringworm, is a common superficial fungal infection in companion animals and humans. Between 2012 and 2023, plucked hair and scraped scale samples from domestic dogs and cats with clinical suspicion of dermatophytosis were collected from 355 veterinary medical centres across mainland Portugal. A total of 4716 animal samples were inoculated onto DERM agar, incubated at 25 °C for up to 4 weeks, and periodically examined macro- and micro-scopically to observe and evaluate fungal growth. Of these, 271 samples were removed due to contaminant fungi. Of the 568 positive cultures, the highest number were from the North (48.1%; 95% CI: 44.0–52.2%) and Centre (32.4%; 95% CI: 28.7–36.4%) regions. *Microsporum canis* was the most frequently isolated species (63.9%), followed by *Trichophyton* spp. (20.3%) and *Nannizia gypsea* (formerly *Microsporum gypseum*) (8.1%). Felines exhibited a higher frequency (17.4%) compared with dogs (9.1%) (*p* < 0.001). In dogs, the Yorkshire Terrier, West Highland White Terrier, Miniature Pinscher, Dalmatian and Miniature Schnauzer demonstrated a significant predisposition to dermatophytosis (*p* < 0.05). In cats, the Persian and Scottish Fold breeds were significantly predisposed (*p* < 0.05). No significant differences were found between sexes (*p* > 0.05). These findings underscore dermatophytosis as an increasing public health concern due to its zoonotic and contagious nature, providing comprehensive insights into the epidemiology of dermatophytosis in Portugal.

## 1. Introduction

Dermatophytes, a group of keratinolytic and keratinophilic fungi, are etiological agents of zoonoses with anthropo–zoonotic transmissions. They cause dermatophytosis, a common skin disease in humans and companion animals, resulting from the superficial fungal infection of keratinised skin structures. This disease manifests as tinea in humans and ringworm in animals [1,2,3,4,5]. The group of dermatophytes comprises 52 keratin-degrading species divided into nine genera: *Trichophyton*, *Microsporum*, *Epidermophyton*, *Arthroderma*, *Lopophyton*, *Nannizia*, *Ctenomyces*, *Guarromyces* and *Paraphyton*, classified by morphology, clinical form and molecular characteristics, and grouped by habitat into anthropophilic, geophilic and zoophilic [6]. Until recently, these fungi had been included in only three genera: *Epidermophyton*, *Microsporum* and *Trichophyton*, which were not monophyletic [7,8,9]. For this reason, and coinciding with the abandonment of the dual nomenclature of fungi in 2013 by the International Code of Nomenclature for algae, fungi, and plants, the number of described genera has increased [10]. Dermatophytosis is an important disease in companion animals due to its pleomorphic clinical presentation, infectious and contagious nature, and zoonotic potential, being particularly prevalent in cats and often caused by the fungus *Microsporum canis* [1,11,12,13,14].

The disease is characterised by multifocal alopecia and crust on the skin with a specific formation, erythema and eventual pruritus [2,7]. The lesions can vary with each animal species [2,7]. In felines and young puppies, the commonly affected areas are the face, ears and muzzle before progressing to other body areas [7]. Transmission occurs via direct contact with another infected host or contaminated fomite [1]. The gold standard diagnostic techniques for the identification of dermatophytosis are based on a combination of physical examination, Wood’s lamp findings and diagnostic testing (direct hair examination, fungal culture) [2,15,16]. PCR can also be a useful diagnostic tool whenever cultures are inconclusive [17]. Treatment includes systemic antifungals such as itraconazole, terbinafine and griseofulvin. Currently, with the emergence of antifungal-resistant isolates, in vitro antifungal susceptibility testing might help to improve the therapy and select an effective antifungal agent against that specific clinical isolate [17].

Dermatophytosis is distributed globally and has been reported in numerous animal species and humans [1,7]. Companion animals can be carriers of some dermatophyte species, which cannot invade the healthy skin of these animals [8,13]. The progression to infection is influenced by certain predisposing factors, such as young age, immunosuppression, nutritional deficiency, high environmental temperature with high humidity or skin trauma [1,8,13]. The prevalence of dermatophytosis has been increasing in companion animals (dogs and cats) and humans, gaining significant attention as a public health problem [1,18]. In Europe, the prevalence of dermatophytosis in dogs and cats ranges from 20% to 30% [7]. Studies have shown that in 81.8% to 97% of the cases, the main species causing dermatophytosis in pets is *M. canis* [1,19]. Worldwide, it has been determined that the prevalence of these infections ranges from 8% to 19% in dogs and 7% to 72% in cats [20].

Although dermatophytosis is commonly reported by veterinary practitioners in Portugal, little is known about the relative importance of the dermatophyte species involved and the differences observed according to animal species, breed, sex and age. Furthermore, as fungal infections are a growing concern due to their increasing frequency and potential association with antimicrobial resistance, the authors assessed the evolution of dermatophytosis diagnosed during the study period.

To the authors’ knowledge, no studies in Portugal have explored the relationships between epidemiological and clinicopathological parameters in dermatophytosis in companion animals, nor have any thoroughly described the epidemiological relationships and their clinical usefulness in dogs and cats. In the present study, the authors present the first epidemiological analysis and its relationship with clinicopathological parameters, marking the first extensive work of its kind conducted in Portugal.

## 2. Materials and Methods

### 2.1. Data Collection, Sampling and Diagnostic Procedures

Samples from suspected cases of dermatophytosis were submitted to INNO Veterinary Laboratories (Braga, Portugal). These samples (*n* = 4716) were collected from 355 veterinary practices, including clinics and hospitals, across all regions of mainland Portugal. Only samples from dogs and cats with alopecia and desquamation reported by veterinary practitioners and consecutively classified as suspected cases of dermatophytosis were used in this study. Each sample included a laboratory requisition with the relevant clinical information such as breed, sex, age, vaccination and prophylactic status, clinical suspicion/clinical signs and requested analyses. The age of the animals was categorised into five groups: puppy/kitten, <1 year old; young, 1 to <2 years old; adult, 2 to <6 years old; senior, 6 to <11 years old; and old, ≥11 years old.

Plucked hairs and/or scraped scales from each animal were collected using a sterile lancet. The sampling site on the animal was disinfected with 70% ethanol and chlorhexidine before collection, and the samples were placed in sterile containers. Wet mount analysis was performed before the culture (Figure 1). Samples were inoculated onto DERM agar plates (bioMérieux, Marcy–l’Étoile, France), incubated at 25 °C for up to 4 weeks and periodically checked for fungal growth. Colony morphology, pigmentation and growth rate of cultures were observed for macroscopic examination. Microscopic examination was performed using glass slides stained with Lactophenol Cotton Blue (Merck, Darmstadt, Germany). Characteristic size, shape, presence of septa, thickness of conidial wall and arrangement of conidial cells around the hyphae were recorded.

*Trichophyton* spp. colonies were observed, presenting as smooth or powdery with colors ranging from white to cream or tan. The reverse side of these colonies exhibited red coloration due to the presence of phenol red in the agar, serving as a pH indicator for dermatophyte positivity (Figure 2).

Microscopic examination of *Trichophyton* spp. revealed numerous microconidia forming dense clusters. These microconidia were hyaline, smooth-walled and predominantly spherical or subspherical. Additionally, some species exhibited cigar-shaped macroconidia with smooth, thin walls. Other microscopic features included spiral hyphae and chlamydospores, which varied among different species (Figure 3).

*Epidermophyton floccosum* forms colonies that quickly change in colour from green to yellow and typically have a velvety texture. Microscopically, this species forms only macroconidia, which are club-shaped with one to five cells. These macroconidia are smooth-walled and arranged singly or in small clusters. *Epidermophyton floccosum* does not form microconidia.

*Microsporum canis* displayed septate hyphae along with macroconidia and microconidia. The macroconidia were spindle-shaped with asymmetrical button-like ends, containing 6–15 compartments and featured long, rough, dense outer walls (Figure 4).

*Nannizzia gypsea* (formerly *Microsporum gypseum*) was characterised by septate hyphae and a significant number of macroconidia and microconidia. The macroconidia were fusiform and symmetrical with rounded ends, containing 3–6 compartments. The microconidia were moderately numerous and located along the hyphae (Figure 5).

*Nannizzia nana* (formerly *Microsporum nanum*) exhibited septate hyphae and pear-shaped macroconidia, typically with two compartments. In this species, microconidia were less common and smaller compared with those observed in other species (Figure 6).

### 2.2. Statistical Analysis

All the data were available in digital format in Clinidata^®^ (Clinidata XXI version 5.3.25, Maxdata Software, S.A., Carregado, Portugal) and transferred to Microsoft Excel^®^ (Microsoft, Redmond, WA, USA) sheets. Statistical analysis was conducted using the JMP^®^, version 14.3 SAS Institute, Cary, NC, USA, 1989–2023 SAS and MedCalc^®^ Statistical Software version 20.006 (MedCalc Software Ltd., Ostend, Belgium, 2021). Non-parametric tests were employed to study the differences between the observed and expected frequencies of categories within a field, including the binomial test, the one-sample Chi-square test and Fisher’s exact test, depending on the number of categories in the categorical field. For comparisons among three or more independent groups, the Kruskal–Wallis test was utilised, followed by the Dunn–Bonferroni post hoc test for multiple comparisons when appropriate. Additionally, logistic regression with Tikhonov regularisation was employed to identify breeds with a higher predisposition for testing positive, accounting for the disproportional representation of breeds in the study. The Cochran–Armitage Trend test was also used to assess trends across ordered categories. The sample parameters were categorised as follows: district, region (NUTS2: Nomenclature of Territorial Units for Statistics), species, breed, sex, age categories and diagnosis (fungal identification)

## 3. Results

### 3.1. Positive Samples and Geographical Distribution

Of the total of 4716 animals included in this study, only 4445 distinct animal samples of plucked hairs and/or scraped scales were analysed, following the removal of contaminant fungi (5.8%; 95% CI: 5.1–6.5%; *n* = 271). Out of 4445 animals, 3877 (87.2%; 95% CI: 86.2–88.2%) tested negative for dermatophyte fungi while 568 (12.8%, 95% CI: 11.8–13.8%) tested positive. The majority of samples originated from Porto (22%; 95% CI: 20.7–23.2%; *n* = 975), Braga (17.8%; 95% CI: 16.7–19.0%; *n* = 792), Lisboa (15.4%; 95% CI: 14.4–16.5%; *n* = 685) and Viseu (9.7%; 95% CI: 8.9–10.6%; *n* = 433) (Figure 7).

The distribution was divided by NUTS2 (Nomenclature of Territorial Units for Statistics) regions of mainland Portugal as follows: 44.6% (95% CI: 43.2–46.1%) North region (*n* = 1983); 28.4% (95% CI: 27.1–29.7%) Centre region (*n* = 1262); 13.9% (95% CI: 12.9–15.0%) Greater Lisbon (GL) (*n* = 619); 4.9% (95% CI: 4.3–5.6%) Península de Setúbal (PdS) (*n* = 218); 3.6% (95% CI: 3.1–4.1%) Oeste e Vale do Tejo (OVT) (*n* = 158); 3.5% (95% CI: 3.0–4.1%) Alentejo (*n* = 157); and 1.1% (95% CI: 0.8–1.4%) Algarve (*n* = 48). The regions with the highest frequency of positive for dermatophyte fungi were the Centre region (14.6%; 95% CI: 12.7–16.6%), followed by the North region (13.8%; 95% CI: 12.3–15.4%).

The Chi-square test revealed a statistically significant association between the region (NUTS2) and the occurrence of dermatophyte fungi (*p* = 0.002). The average frequency of dermatophyte in this study, across different regions, was 12.8% (95% CI: 11.8–13.8%), with an average percentage variation between 8.3% (ranging from 6.3% to 14.6%). No statistical correlation was observed between dermatophyte fungi isolations across districts and regions (NUTS2). Table 1 illustrates the occurrence of dermatophyte fungi isolation in the current study from the years 2012 to 2023 across the geographical regions of mainland Portugal.

Table 2 illustrates the percentage of positive dermatophyte fungi isolations in the current study between 2013 and 2023 across the regions of mainland Portugal, excluding 2012, for which all samples yielded contaminant fungi.

Figure 8 displays the average percentage of positivity for dermatophyte fungi over a 12-year period in mainland Portugal (2012–2023).

Table 3 summarises the distribution of positive dermatophyte isolations (*n* = 568) across the regions of mainland Portugal from 2012 to 2023. The North region had the highest frequency of cases, accounting for 48.1% (95% CI: 44.0–52.2%; *n* = 273) of the total diagnoses. The Centre region followed with 32.4% (95% CI: 28.7–36.4%; *n* = 184), and Greater Lisbon reported 11.4% (95% CI: 9.1–14.3%; *n* = 65). *Epidermophyton floccosum* was predominantly found in the North (44.4%; 95% CI: 18.9–73.3%; *n* = 4) and Centre (33.3%; 95% CI: 12.1–64.6%; *n* = 3). *Microsporum canis* was the most frequent species, especially in the North (49.6%; 95% CI: 44.5–54.7%; *n* = 180), and *Nannizzia gypsea* (formerly *Microsporum gypseum*) had the highest occurrence in the North as well (65.2%; 95% CI: 50.8–77.3%; *n* = 30). *Nannizzia nana* (formerly *Microsporum nanum*) was primarily identified in the North (34.3%; 95% CI: 20.8–50.8%; *n* = 12) and Centre (31.4%; 95% CI: 18.6–48.0%; *n* = 11). *Trichophyton* spp. were most commonly diagnosed in the North (40.9%; 95% CI: 32.3–50.0%; *n* = 47). The other regions exhibited lower percentages of positive diagnoses.

### 3.2. Animal Species

From the 4445 animals analysed, 2478 (55.8%) were canine and 1967 (44.2%) were feline. Table 4 represents the percentage of positive and negative according to species. The Mann–Whitney U-Test revealed significant differences between the groups (U = 2,235,646, z = −8.2, *p* < 0.001). The mean rank for the canine group was 2141.7, whereas the mean rank for the feline group was 2325.42. The effect size was 0.12, indicating a small but statistically significant difference at the 0.05 significance level. Thus, it was concluded that the differences observed between species were significant (*p* < 0.001), with felines being more affected by dermatophytosis than canines.

For canines, 2252 (90.9%) out of 2478 tested negative for dermatophytosis, while 226 (9.1%) tested positive. In felines, 1625 (82.6%) out of 1967 were negative, and 342 (17.4%) were positive. Overall, out of the total 4445 animals, 3877 (87.2%) tested negative, and 568 (12.8%) tested positive for dermatophytosis.

This study highlights a higher frequency of dermatophytosis in felines (17.4%) compared with canines (9.1%).

### 3.3. Animal Breed

Regarding canine breeds, our study comprised dogs from 86 different breeds, including 785 mixed-breed dogs (17.7%), 314 Labrador Retrievers (7.1%), 182 French Bulldogs (4.1%), 148 Pinschers (3.3%), 98 German Shepherds (2.2%), 94 Yorkshire Terriers (2.1%), 48 Shar Peis (1.1%), 46 Golden Retrievers (1%), 43 Boxers (1%), 38 English Bulldogs (0.9%) and 76 other breeds.

Regarding feline breeds, our study comprised cats from 13 different breeds, including 1675 Domestic Shorthairs (85.2%), 156 Persians (7.9%), 64 Siamese (3.3%), 25 Scottish Folds (1.3%), 13 British Shorthairs (0.7%), 10 Norwegian Forests (0.5%), 7 Bengals (0.4%), 5 Maine Coons (0.3%), 3 Scottish Straight Folds (0.2%) and 4 other breeds.

For statistical analysis, the canine mixed-breed and feline domestic shorthair categories were excluded.

The Kruskal–Wallis test among canine breeds did not indicate significant associations (*p* = 0.112), suggesting a uniform frequency of dermatophytosis across the breeds examined. Detailed ranks and pairwise comparisons were conducted, revealing specific breed differences; however, none achieved significance after adjusting for multiple comparisons. Due to the inconclusive results of these tests, a logistic regression analysis with Tikhonov regularisation was conducted to determine if there was a predisposition among various canine breeds to dermatophytosis. The results indicated that the majority of breeds did not show a statistically significant association with the condition (*p* > 0.05). However, notable exceptions included the Yorkshire Terrier (*p* < 0.001), West Highland White Terrier (*p* = 0.021), Miniature Pinscher (*p* = 0.005), Dalmatian (*p* = 0.039) and Miniature Schnauzer (*p* = 0.019), which exhibited significant positive associations. Consequently, while most breeds did not exhibit a heightened vulnerability to dermatophytosis, these specific breeds were identified as having a statistically significant predisposition. The model demonstrated a high specificity of 99.9% and an overall accuracy of 90.9%.

In contrast, among feline breeds, the Kruskal–Wallis test indicated a significant association (*p* = 0.045), suggesting variability in dermatophytosis frequency across breeds. Detailed ranks and pairwise comparisons were performed, revealing specific breed differences, but none reached significance after adjustment for multiple comparisons. The Dunn–Bonferroni post hoc test revealed no significant associations, with all adjusted *p*-values exceeding the 0.05 significance level. Although the overall test suggested differences among breeds, individual comparisons did not show significant variation in dermatophytosis frequency after correction for multiple testing. Subsequently, a logistic regression analysis with Tikhonov regularisation was conducted to determine if there was a predisposition among various feline breeds to dermatophytosis, indicating that most feline breeds did not have a statistically significant association with the condition (*p* > 0.05). Notable exceptions included the Persian (*p* < 0.001) and Scottish Fold (*p* = 0.011), which showed significant positive associations. Consequently, while the majority of breeds did not exhibit a heightened vulnerability to dermatophytosis, these specific breeds were identified as having a statistically significant predisposition. The model demonstrated a high specificity of 99.9% and an overall accuracy of 82.7%.

### 3.4. Sex

From the 2478 canines analysed, 1119 (45.2%; 95% CI: 43.2–47.1%) were females, and 1359 (54.8%; 95% CI: 52.9–56.8%) were males. Regarding felines, of the 1967 analysed, 949 (48.2%; 95% CI: 46.0–50.5%) were females, and 1017 (51.8%; 95% CI: 49.5–54.0%) were males. Table 5 represents the percentage of positive and negative according to sex. The differences observed between animal sexes in both canine and feline species were not significant (*p* > 0.05). Table 5 displays the occurrence of dermatophyte fungi isolation according to sex in canine and feline samples.

### 3.5. Age

From the 4445 animals that were analysed, age data were available for only 3790 animals because, for 655 animals (14.7%), requisition documents did not specify their age and were thus excluded from certain analytical processes. The age distribution among these 3790 animals ranged from ≤1 year (6 months) to 20 years in both species, with an average age of 6.6 ± 6.43 years in dogs and 6.5 ± 6.42 years in cats. In dogs, 24.6% (95% CI: 22.8–26.4%; *n* = 524) were puppies, 6.2% (95% CI: 5.2–7.3%; *n* = 132) were young, 34.7% (95% CI: 32.7–36.8%; *n* = 741) were adults, 26.3% (95% CI: 24.4–28.2%; *n* = 560) were seniors and 8.3% (95% CI: 7.2–9.5%; *n* = 176) were old. In cats, 32.0% (95% CI: 29.8–34.3%; *n* = 530) were kittens, 5.1% (95% CI: 4.2–6.3%; *n* = 85) were young, 35.0% (95% CI: 32.4–37.0%; *n* = 574) were adults, 20.5% (95% CI: 18.6–22.5%; *n* = 340) were seniors and 7.7% (95% CI: 6.5–9.1%; *n* = 128) were old.

Table 6 displays the occurrence of dermatophyte fungi isolation according to age group. The result of the Spearman correlation analyses for both canine and feline populations reveal significant relationships between age groups and the frequency of dermatophytosis. In dogs, the correlation coefficient (r) is −0.05 (*p* = 0.018), indicating a negligible, negative correlation, meaning that as the age group increases, the frequency of dermatophytosis slightly decreases. In cats, the correlation coefficient (r) is −0.14 (*p* < 0.001), showing a low, negative correlation, suggesting that dermatophytosis decreases more noticeably with increasing age. Both analyses show a minor yet significant inverse relationship between age groups and the occurrence of dermatophytosis in dogs and cats, with *p*-values below 0.05 confirming statistical significance.

### 3.6. Dermatophyte Diagnosis

The analysis reveals a significant association between species (canine and feline) and dermatophyte genera/species, confirmed by Pearson’s likelihood ratio (31.508, *p* < 0.001) and Fisher’s Exact test (*p* < 0.001). *Microsporum canis* is significantly more frequent in felines (68.6%; 95% CI: 63.6–73.2%) compared with canines (31.4%; 95% CI: 26.8–36.4%), while *Trichophyton* spp. are more frequent in canines (52.2%; 95% CI: 43.1–61.1%) than in felines (47.8%; 95% CI: 38.9–56.9%). The Cochran–Armitage Trend test supports these findings, indicating a significant trend (Z = 4.512232, *p* < 0.001). Consequently, the species of the animal significantly influences the distribution of different dermatophyte infections, as detailed in Table 7.

*Epidermophyton floccosum* was detected in 0.7% of felines and 0.9% of canines (1.6%; *n* = 9). *Microsporum canis* was found in 43.3% of felines and 19.7% of canines, with 363 cases (63.9%) overall. *Nannizzia gypsea* appeared in 3.9% of felines and 4.2% of canines, amounting to 46 cases (8.1%). *Nannizzia nana* was identified in 2.1% of felines and 4.1% of canines, summing up to 35 cases (6.2%). *Trichophyton* spp. were present in 9.7% of felines and 10.6% of canines (20.3%; *n* = 115).

This study demonstrates that *M. canis* is significantly more frequent in felines. Out of the total 568 dermatophyte cases analysed, felines represented 60.2% and canines 39.8%.

## 4. Discussion

### 4.1. Descript Data and Geographical Distribution

In the present study, 12.8% of the analysed animals tested positive for dermatophyte fungi, highlighting the frequency of dermatophytosis in companion animals across mainland Portugal. In a parallel study focusing on shelters in North Portugal, a lower prevalence of 3.5% was reported [21]. This discrepancy in frequency rates may originate from differences in study populations, sampling methods or regional variations. In other studies [1,19,21], *M. canis* was the predominant species identified, which aligns with the present study’s findings that *M. canis* constitutes 63.9% of positive samples. Our study diverges from studies in other countries [11,13,15,22,23], indicating an extensive geographical distribution. The highest frequencies were observed in the North (48.1%) and Centre (32.4%) regions, which may be attributed to the varying climatic conditions, especially higher humidity levels in these areas. Dermatophytes are more commonly found in humid environments and urban areas [24,25,26,27]. While the frequency is significantly lower in the driest regions of the south, such as Alentejo (2.5%) and Algarve (0.5%), these findings still demonstrate that dermatophytosis exists throughout mainland Portugal, contrary to the belief that dermatophytosis is primarily confined to humid areas in the Centre and North of Portugal. The southern regions also exhibit relatively high-frequency rates: Greater Lisbon (11.4%), Península de Setúbal (2.8%), Alentejo (2.5%) and Algarve (0.5%), demonstrating that no geographical area in mainland Portugal is free from dermatophytosis.

This broader distribution in Portugal is consistent with epidemiological studies from other countries, suggesting that the spread of dermatophytes may be influenced by climatic changes [22,28]. The significant correlation between the geographical region and dermatophyte infection outcomes (*p* < 0.001) underscores the impact of location on disease frequency.

#### Dermatophyte Isolation Trends in Mainland Portugal

The frequency of dermatophyte isolation in mainland Portugal from 2013 to 2023 exhibits diverse trends across various regions. An analysis of the data reveals that most regions, including the Centre, Oeste e Vale do Tejo (OVT), Península de Setúbal (PdS) and Alentejo, demonstrate an increasing trend, with Oeste e Vale do Tejo (OVT) experiencing a particularly marked rise. Conversely, the North and Algarve regions show a decreasing trend in dermatophyte isolation percentages. The Greater Lisbon (GL) area remains relatively stable, with a slight upward trend. These trends underscore the significance of regional monitoring and the implementation of tailored public health strategies to address the specific needs of each area. Overall, despite a decline in some regions, the general trend indicates an apparent increasing frequency of dermatophyte isolation, especially in recent years.

### 4.2. Species

This study highlights a higher frequency of dermatophytosis in felines (17.4%) compared with canines (9.1%); however, this may reflect a sampling bias, as clinical diagnosis in felines might be more accurate, resulting in more true positive samples, whereas canines might have more conditions that mimic dermatophytosis, leading to a higher number of negative tests.

Our findings are consistent with other studies that have shown a higher susceptibility of cats to dermatophyte infections [11,12,13,14,21]. Felines, particularly those in environments with high population densities, such as shelters and catteries, are more prone to contracting and spreading dermatophytosis. This increased frequency in felines can be attributed to several factors, including their grooming habits, which can facilitate the spread of spores across their fur, and their closer contact with potentially contaminated environments [29,30].

Moreover, the data underscores the need for targeted preventive measures in feline populations, given the zoonotic potential of dermatophytosis. Strategies such as regular screening, improved hygiene practices in shelters and prompt treatment of infected animals are essential in managing and reducing the incidence of dermatophytosis.

### 4.3. Breed

This study analysed dermatophytosis frequency across 86 canine breeds and 13 feline breeds in Portugal. Mixed-breed dogs and feline domestic shorthairs were excluded from specific analysis due to their high heterogeneity, which could introduce significant variability and mask breed-specific trends. This exclusion was intended to ensure the clarity and specificity of our breed-related findings. In dogs, mixed-breeds constituted the largest group (17.7%), followed by Labrador Retrievers (7.1%) and French Bulldogs (4.1%). The Kruskal–Wallis test did not show significant differences among breeds (*p* = 0.112), suggesting a uniform frequency across most breeds. However, logistic regression identified certain breeds, such as Yorkshire Terriers, West Highland White Terriers, Miniature Pinschers, Dalmatians and Miniature Schnauzers, as having a significantly higher predisposition to dermatophytosis. This indicates that specific breeds may be more vulnerable due to genetic or behavioural factors, which align with the findings from other authors [31].

In contrast, the feline analysis showed a significant association between breed and dermatophytosis frequency (*p* < 0.05), with Domestic Shorthairs being the most common (85.2%). Despite the overall significant association, detailed comparisons did not show significant differences after adjusting for multiple comparisons. Logistic regression revealed that Persians and Scottish Folds have a significantly higher predisposition to the infection. This suggests that certain feline breeds are more susceptible, possibly due to breed-specific characteristics such as grooming habits or coat type, which aligns with findings from other studies indicating that long-haired cats are more susceptible to dermatophytosis compared with short-haired cats, with a prevalence of 34.9% in long-haired cats versus 6.3% in short-haired cats [29].

These findings highlight the importance of breed-specific approaches in managing dermatophytosis. For breeds with higher susceptibility, enhanced monitoring, regular screenings and targeted treatments are recommended. Understanding the genetic and environmental factors contributing to breed predisposition can improve prevention and control strategies, particularly in high-risk environments such as shelters and catteries.

### 4.4. Sex

The distribution of positive dermatophyte diagnoses between sexes in this study shows no significant difference in infection rates (*p* > 0.05), which contrasts with another study [21] that observed a significant prevalence (*p* = 0.027) in females (5.5%) compared with males (1.3%). Among the 2478 canines analysed, 45.2% were female, and 54.8% were male. For felines, among the 1967 samples, 48.3% were female, and 51.8% were male. Despite the apparent numerical differences, statistical analysis revealed that these variations were not significant (*p* > 0.05).

This finding suggests that contrary to certain diseases where sex might influence susceptibility due to hormonal, genetic or behavioural factors, dermatophytosis in this population does not significantly differ between males and females. Consequently, preventive measures and clinical interventions for dermatophytosis should be uniformly applied across both sexes, with a greater focus on other risk factors such as age, breed and environmental conditions.

The lack of significant sex-based differences in dermatophytosis aligns with existing literature [22,32], although some studies have reported that male dogs are more affected by dermatophytes [28].

### 4.5. Age

The age distribution analysis of the 3790 animals revealed that younger animals, particularly puppies (13.8%) and kittens (14%), had higher susceptibility to dermatophytosis, with a decline in frequency as animals aged. The correlation analysis showed a significant inverse relationship between age and dermatophytosis frequency in both dogs (r = −0.05, *p* = 0.018) and cats (r = −0.14, *p* < 0.001), indicating that as animals grow older, the likelihood of infection decreases. This pattern is consistent with previous studies [15,21,23,28], reinforcing that age is a critical factor in the susceptibility to dermatophytosis.

These findings highlight the importance of focusing preventive measures and surveillance on younger animals, who are at a higher risk for dermatophytosis, and that targeted interventions could be directed toward puppies and kittens.

### 4.6. Dermatophyte Diagnosis

The study reveals a significant species-specific distribution of dermatophyte infections, with distinct patterns observed between canines and felines. *Microsporum canis* was found to be significantly more frequent in felines, with 43.8% of infected cats compared with 20.1% of infected dogs. This indicates a higher susceptibility of cats to this particular dermatophyte, which may be due to factors such as their grooming habits, which can spread spores more effectively across their bodies, and their close contact with potentially contaminated environments [2].

In contrast, *Trichophyton* spp. were more commonly found in dogs (10.6%) than in cats (9.7%). This suggests that canines may encounter different environmental or behavioural risk factors that make them more prone to this type of dermatophyte. For instance, dogs’ outdoor activities and interaction with soil and other animals might increase their exposure to *Trichophyton* spp.

These results emphasise the importance of considering the species when diagnosing and treating dermatophytosis, a pattern that is consistent with previous studies [20,21]. For felines, where *Microsporum canis* predominates, preventive measures might include minimising environmental contamination and managing grooming habits. For canines, strategies might focus on reducing outdoor exposure and improving hygiene in areas frequented by dogs [15,21,29,30].

This study offers valuable insights into the epidemiology of dermatophytosis in dogs and cats in Portugal, underscoring the necessity for continuous monitoring and effective control measures to manage the public health risks associated with this zoonotic disease, especially in environments with potentially poor hygiene, such as shelters or catteries, which significantly increase the risk of dermatophytosis infection.

## 5. Conclusions

This study provides a comprehensive analysis of dermatophytosis in companion animals across mainland Portugal over a twelve-year period, highlighting the significant frequency of *Microsporum canis* among domestic cats and dogs. The findings indicate a higher susceptibility of felines compared with canines, underscoring the necessity for species-specific prevention and treatment strategies, especially in high-density environments like shelters and catteries. For dogs, minimising exposure to contaminated environments and ensuring good hygiene practices are recommended. For cats, particularly those with long hair, a clean living environment and strategies addressing grooming habits are essential due to their increased risk for spore retention.

Although dermatophyte infections do not show significant differences in infection rates between sexes, they do exhibit significant age-related differences, with puppies and kittens being particularly vulnerable. Additionally, notable regional differences in dermatophytosis frequency have been identified, with the highest levels observed in the North and Centre regions of Portugal, which are the more humid regions. These geographical variations, combined with notable associations between specific breeds and susceptibility—particularly toy and small breed dogs, as well as Persians and Scottish Folds in cats—suggest that environmental factors and breed-specific traits significantly influence the distribution of infections.

Over the twelve-year period from 2012 to 2023, this study shows diverse trends in dermatophyte isolation across different regions. Most regions, including the Centre, Oeste e Vale do Tejo (OVT), Península de Setúbal (PdS) and Alentejo, demonstrate an increasing trend, while the North and Algarve regions show a decreasing trend. Overall, despite a decline in some regions, the general trend indicates an apparent increasing frequency of dermatophyte isolation, especially in recent years.

The identification of multiple dermatophyte species, such as *Microsporum canis* and *Nannizzia gypsea*, provides crucial epidemiological data and insights into clinical management strategies. Geophilic species like *N. gypsea*, which primarily reside in the soil, require specific environmental control measures to prevent infection, while zoophilic species like *M. canis* demand targeted treatment approaches. These findings enhance our understanding of the distribution, pathogenicity and treatment of dermatophyte infections, ultimately contributing to better health outcomes for both animals and humans.

In summary, this research elucidates the complex epidemiological patterns of dermatophytosis in companion animals, emphasising the necessity for adequate diagnostic, surveillance and disease management approaches. These insights significantly deepen our understanding of dermatophytosis within Portugal and advocate for the importance of the One Health approach, a cornerstone of the Stockholm Paradigm in Planetary Health. By understanding and addressing the specific risk factors and environmental conditions that contribute to infection, more effective control measures can be implemented to reduce the prevalence and impact of this zoonotic disease.

## Figures and Tables

**Figure 1 microorganisms-12-01727-f001:**
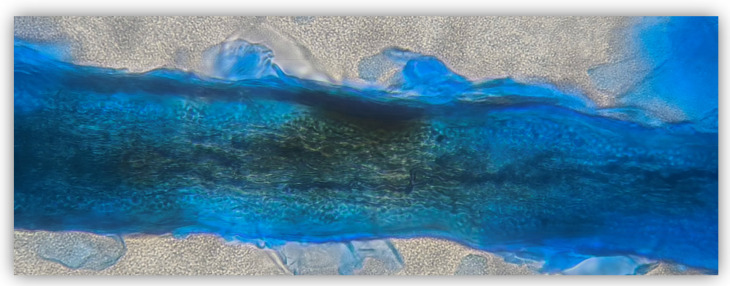
Dermatophytosis (ectothrix) in a cat. Arthrospores on the exterior of the hair shaft (Lactophenol Cotton Blue stain, 400×).

**Figure 2 microorganisms-12-01727-f002:**
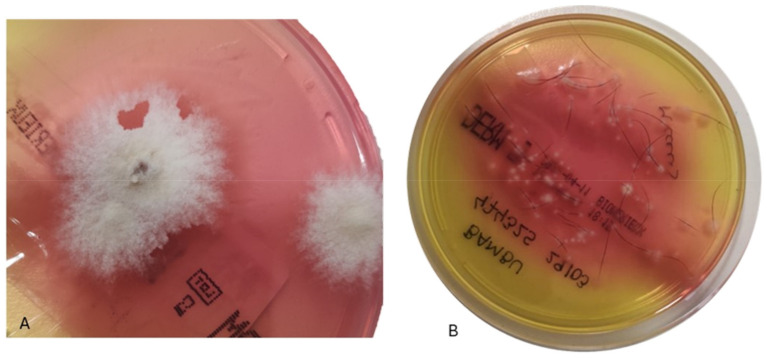
Samples were inoculated onto DERM agar plates (bioMérieux, Marcy–l’Étoile, France), incubated at 25 °C. *Trichophyton* spp. with colonies smooth or powdery, with colours ranging from white to cream or tan (**A**). The reverse side of these colonies can show the colour red (**B**) due to the phenol red in the agar as a pH indicator when dermatophyte positive.

**Figure 3 microorganisms-12-01727-f003:**
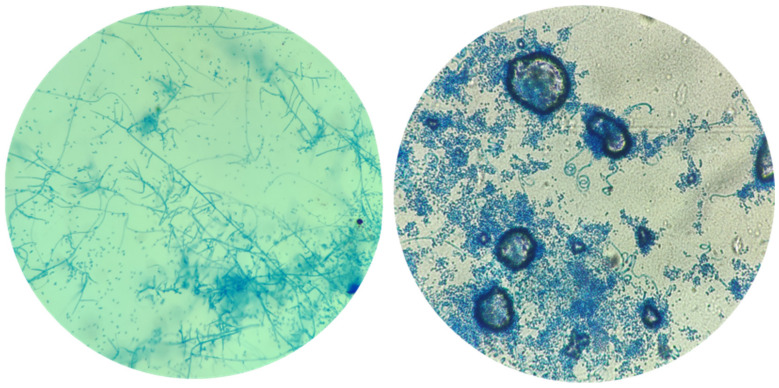
*Trichophyton* spp. typically have numerous microconidia forming dense clusters. These microconidia are hyaline, smooth-walled and predominantly spherical or subspherical. Some species also exhibit cigar-shaped macroconidia with smooth, thin walls. Additional microscopic features can include spiral hyphae and chlamydospores, varying among different species (Lactophenol Cotton Blue stain, 1000×).

**Figure 4 microorganisms-12-01727-f004:**
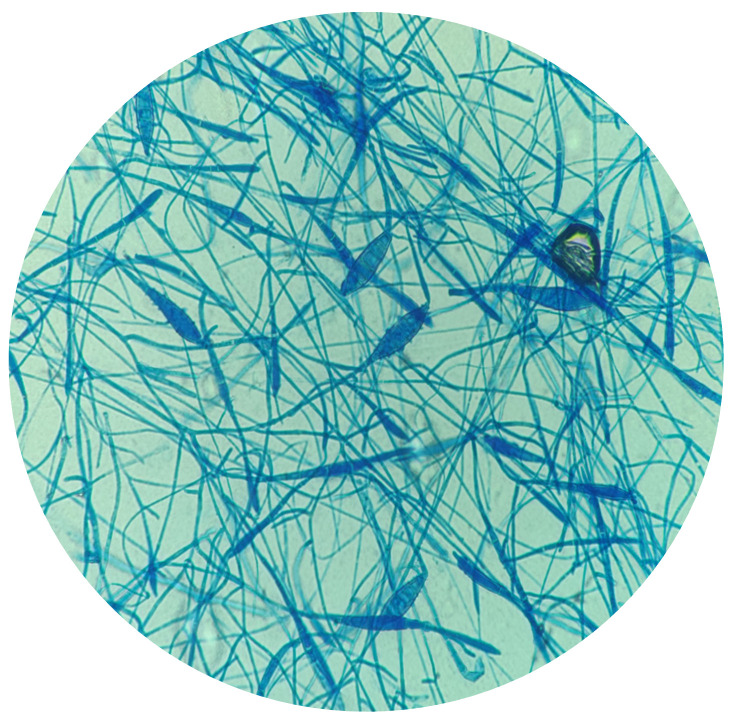
*Microsporum canis* presents septate hyphae, spindle-shaped macroconidia with 6–15 compartments and asymmetrical ends and microconidia. The macroconidia are long, rough and have dense outer walls (Lactophenol Cotton Blue stain, 1000×).

**Figure 5 microorganisms-12-01727-f005:**
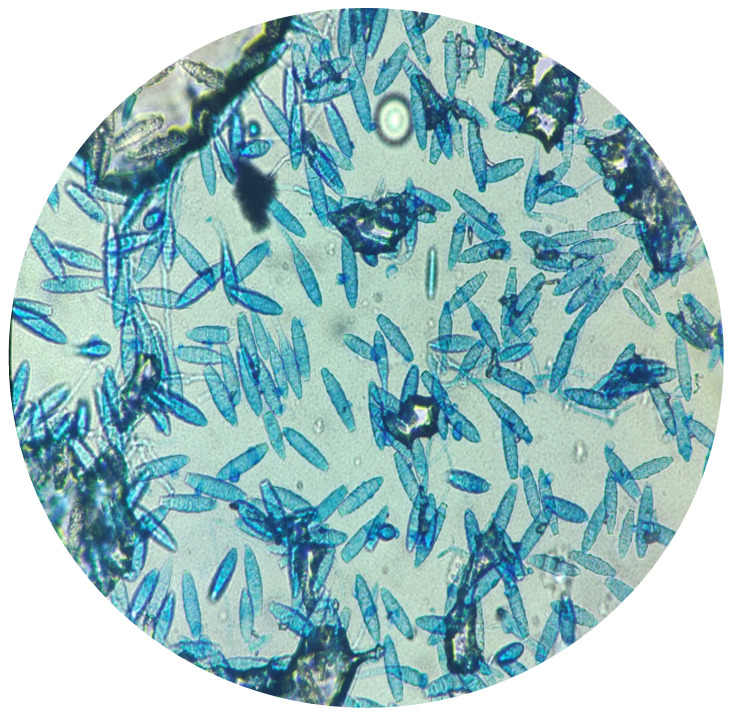
*Nannizzia gypsea* (formerly *Microsporum gypseum*) shows septate hyphae and a significant number of macroconidia and microconidia. The macroconidia are fusiform and symmetrical with rounded ends, containing 3–6 compartments. Microconidia are moderately numerous and are located along the hyphae (Lactophenol Cotton Blue stain, 1000×).

**Figure 6 microorganisms-12-01727-f006:**
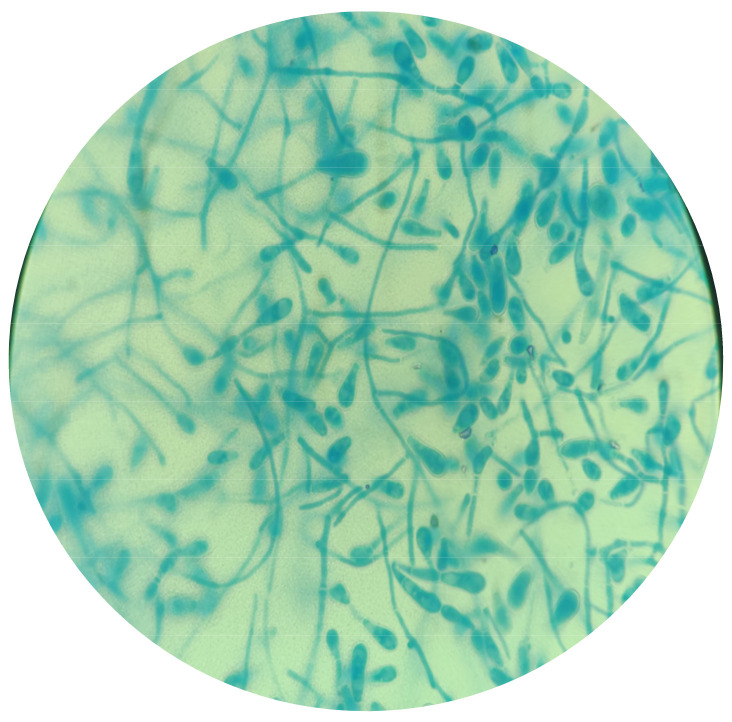
*Nannizzia nana* (formerly *Microsporum nanum*) exhibits septate hyphae and pear-shaped macroconidia, usually with two compartments. Microconidia are less common and smaller compared to those in other species (Lactophenol Cotton Blue stain, 1000×).

**Figure 7 microorganisms-12-01727-f007:**
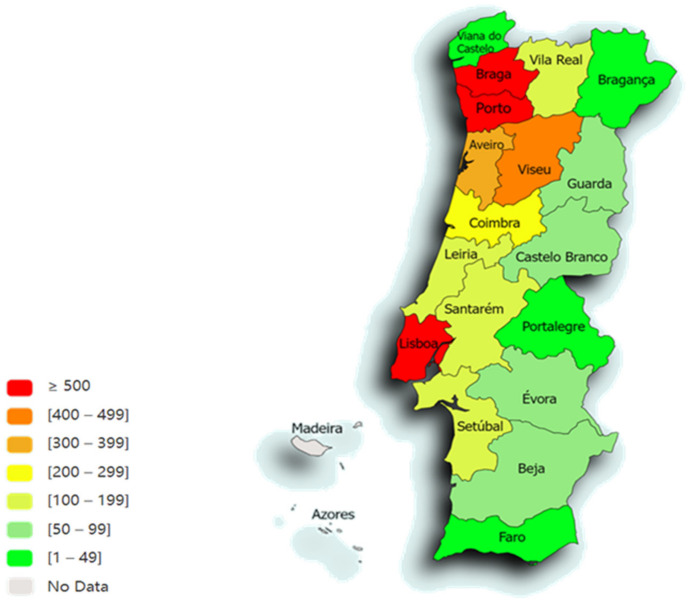
Spatial distribution, according to the distribution of animals from the different districts of Portugal, of the 4445 animals included in this study (map drawn in paintmaps.com; accessed on 23 June 2024).

**Figure 8 microorganisms-12-01727-f008:**
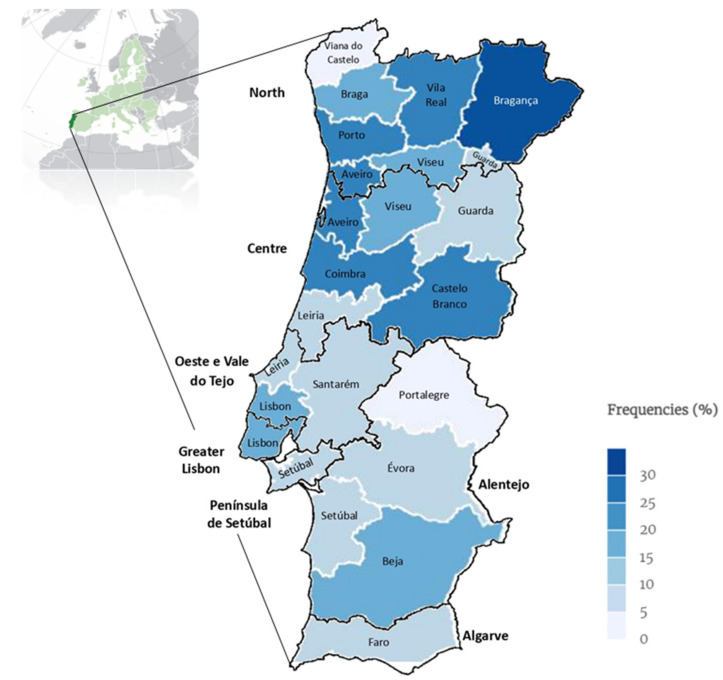
Map of continental Portugal showing a categorical representation of average frequencies of dermatophyte diagnosis over 12 years (2012–2023) per district and region (NUTS2) (map drawn in mapinseconds.com; accessed on 23 June 2024).

**Table 1 microorganisms-12-01727-t001:** Occurrence of dermatophyte fungi isolation by regions (NUTS2) in the 4445 animals included in this study.

		Occurrence of Dermatophyte Fungi				
		Negative		Positive		Total
		*n*	% within Regions	*n*	% within Regions (Frequency)	*n*
Regions (NUTS2)	North	1710 (38.5%)	44.1%	273 (6.1%)	48.1%	1983 (44.6%)
	Centre	1078 (24.3%)	27.8%	184 (4.1%)	32.4%	1262 (28.4%)
	Oeste e Vale do Tejo (OVT)	145 (3.3%)	3.7%	13 (0.3%)	2.3%	158 (3.6%)
	Greater Lisbon (GL)	554 (12.5%)	14.3%	65 (1.5%)	11.4%	619 (13.9%)
	Península de Setúbal (PdS)	202 (4.5%)	5.2%	16 (0.4%)	2.8%	218 (4.9%)
	Alentejo	143 (3.2%)	3.7%	14 (0.3%)	2.5%	157 (3.5%)
	Algarve	45 (1%)	1.2%	3 (0.1%)	0.5%	48 (1.1%)
	Total	3877 (87.2%)	100%	568 (12.8%)	100%	4445 (100%)

OVT, Oeste e Vale do Tejo; GL, Greater Lisbon; PdS, Península de Setúbal; NUTS2, Nomenclature of Territorial Units for Statistics.

**Table 2 microorganisms-12-01727-t002:** Yearly rate trends in dermatophyte fungi isolation within each region across mainland Portugal (2013–2023).

	Evolution of the Percentage of Dermatophyte Isolation over 11 Years in Mainland Portugal											
Regions	2013	2014	2015	2016	2017	2018	2019	2020	2021	2022	2023	Mean
North	26.9%	23.0%	9.0%	10.2%	13.1%	8.4%	14.4%	7.7%	11.4%	17.7%	21.0%	14.8%
Centre	0.0%	31.0%	10.1%	1.6%	17.9%	9.9%	13.0%	10.5%	13.6%	13.6%	24.0%	13.2%
OVT	0.0%	0.0%	11.1%	10.5%	7.4%	0.0%	7.7%	0.0%	15.4%	11.1%	42.9%	9.6%
GL	33.3%	0.0%	0.0%	7.7%	0.0%	0.0%	8.2%	4.0%	7.1%	16.7%	15.8%	8.4%
PdS	0.0%	0.0%	0.0%	16.7%	0.0%	3.3%	11.1%	0.0%	8.0%	10.0%	17.4%	6.0%
Alentejo	0.0%	0.0%	0.0%	25.0%	5.9%	15.4%	11.5%	3.3%	4.8%	12.0%	20.0%	8.9%
Algarve	0.0%	16.7%	0.0%	0.0%	33.3%	0.0%	0.0%	0.0%	0.0%	16.7%	0.0%	6.1%
Mean	8.6%	10.1%	4.3%	10.2%	11.1%	5.3%	9.4%	3.6%	8.6%	14.0%	20.2%	9.6%

OVT, Oeste e Vale do Tejo; GL, Greater Lisbon; PdS, Península de Setúbal.

**Table 3 microorganisms-12-01727-t003:** Distribution of dermatophyte isolation by region and species in mainland Portugal (2012–2023).

		*Epidermophyton floccosum*		*Microsporum canis*		*Nannizzia gypsea* (Formerly *M. gypseum*)		*Nannizzia nana* (Formerly *M. nanum*)		*Trichophyton* spp.		Total
		*n*	% within Region	*n*	% within Region	*n*	% within Region	*n*	% within Region	*n*	% within Region	*n*
Regions (NUTS2)	North	4	44.4%	180	49.6%	30	65.2%	12	34.3%	47	40.9%	273
	Centre	3	33.3%	118	32.5%	12	26.1%	11	31.4%	40	34.8%	184
	OVT	0	0.0%	9	2.5%	0	0.0%	1	2.9%	3	2.6%	13
	GL	2	22.2%	37	10.2%	3	6.5%	8	22.9%	15	13.0%	65
	PdS	0	0.0%	10	2.8%	0	0.0%	2	5.7%	4	3.5%	16
	Alentejo	0	0.0%	7	1.9%	1	2.2%	1	2.9%	5	4.4%	14
	Algarve	0	0.0%	2	0.6%	0	0.0%	0	0.0%	1	0.9%	3
	Total	9	100%	363	100%	46	100%	35	100%	115	100%	568

OVT, Oeste e Vale do Tejo; GL, Greater Lisbon; PdS, Península de Setúbal; NUTS2, Nomenclature of Territorial Units for Statistics.

**Table 4 microorganisms-12-01727-t004:** Occurrence of dermatophyte fungi isolation by species in the 4445 animals included in this study.

Dermatophyte Isolation					
Species	Negative	% within Species	Positive	% within Species	Total
Canine	2252 (50.7%)	90.9%	226 (5.1%)	9.1%	2478 (55.8%)
Feline	1625 (36.6%)	82.6%	342 (7.7%)	17.4%	1967 (44.2%)
Total	3877 (87.2%)	–	568 (12.8%)	–	4445 (100%)

**Table 5 microorganisms-12-01727-t005:** Occurrence of dermatophyte fungi isolation by sex in the 4445 animals included in this study.

		Dermatophyte Isolation		
	Sex	Negative	Positive	Total
Canine	Female	1016 (90.8%)	103 (9.2%)	1119 (45.2%)
	Male	1236 (91.0%)	123 (9.1%)	1359 (54.8%)
Feline	Female	796 (83.9%)	153 (16.1%)	949 (48.3%)
	Male	829 (81.4%)	189 (18.6%)	1018 (51.8%)
	Total	3877 (87.2%)	568 (12.8%)	4445 (100%)

**Table 6 microorganisms-12-01727-t006:** Occurrence of dermatophyte fungi isolation by age in the 3790 animals included in this study.

Age Group							
		Puppy/Kitten (<1 Year Old)	Young (1 to <2 Years Old)	Adult (2 to <6 Years Old)	Senior (6 to <11 Years Old)	Old (≥11 Years Old)	Total
Canine	Negative	456 (21.4%)	122 (5.7%)	684 (32.1%)	511 (24.0%)	162 (7.6%)	1935 (90.7%)
	Positive	68 (3.2%)	10 (0.5%)	57 (2.7%)	49 (2.3%)	14 (0.7%)	198 (9.3%)
Feline	Negative	375 (22.6%)	71 (4.3%)	509 (30.7%)	305 (18.4%)	104 (6.3%)	1364 (82.3%)
	Positive	155 (9.4%)	14 (0.8%)	65 (3.9%)	35 (2.1%)	24 (1.5%)	293 (17.7%)
	Total	1054 (27.8%)	217 (5.7%)	1315 (34.7%)	900 (23.8%)	304 (8.0%)	3790 (100%)

**Table 7 microorganisms-12-01727-t007:** Distribution of dermatophyte by species (feline and canine) in the 3790 animals included in this study.

		Species				
		Feline		Canine		Total
		*n*	% within Species	*n*	% within Species	*n*
Dermatophyte	*Epidermophyton floccosum*	4 (0.7%)	44.4%	5 (0.9%)	55.6%	9 (1.6%)
	*Microsporum canis*	249 (43.3%)	68.6%	114 (19.7%)	31.4%	363 (63.9%)
	*Nannizzia gypsea* (formely *Microsporum gypseum)*	22 (3.9%)	47.8%	24 (4.2%)	52.2%	46 (8.1%)
	*Nannizzia nana* (formely *Microsporum nanum)*	12 (2.1%)	34.3%	23 (4.1%)	65.7%	35 (6.2%)
	*Trichophyton* spp.	55 (9.7%)	47.8%	60 (10.6%)	52.2%	115 (20.3%)
	Total	342 (60.2%)		226 (39.8%)		568 (100%)

## Data Availability

The data presented in this study are available upon request from the corresponding authors.

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
