# Peer review of "Dermatophytosis in Companion Animals in Portugal: A Comprehensive Epidemiological Retrospective Study of 12 Years (2012–2023)"

_microorganisms, 2024, doi:10.3390/microorganisms12081727_

Round 1

Reviewer 1 Report

Comments and Suggestions for Authors

Dear authors,

Congratulations for your work. There are no comments on the paper, the research design is OK, the methods used, especially the statistical analysis, are adequate and the results are quite interesting. 

I have a suggestion: please, add more European contributions on this subject, with some discussion on the differences noticed.

Best regards!

Author Response

Comments 1: Dear authors,

Congratulations for your work. There are no comments on the paper, the research design is OK, the methods used, especially the statistical analysis, are adequate and the results are quite interesting. 

I have a suggestion: please, add more European contributions on this subject, with some discussion on the differences noticed.

Best regards!

Authors’ response (AR) 1: Thank you very much for your kind words and positive feedback on our work. We are pleased to hear that you found our research design, methods, and results to be satisfactory.

In response to the Reviewer’s suggestion, we have added more European contributions to our discussion. In particular, we have included the following reference, which provides valuable insights into dermatophyte carriage in shelter and clinic cats and dogs of northern Portugal:

  • Afonso, P.; Quintas, H.; Vieira, A.; Pinto, E.; Matos, M.; Soares, A.; Cardoso, L.; Coelho, A.C. Furry Hosts and Fungal Guests: Investigating Dermatophyte Carriage in Shelter and Clinic Cats and Dogs of Northern Portugal. Vet Glas 2024, 78, 28–46. https://doi.org/10.2298/VETGL240130006A

Reviewer 2 Report

Comments and Suggestions for Authors

Address these comments:

ABSTRACT

24: 4445 animals were sampled in the main text whereas 4716 is stated in the abstract

26: Grammar check

27: Centre or central

Why was there no antifungal susceptibility testing in the study?

The conclusion of the abstract is lacking in that it did not show whether the trend showed increasing or decreasing incidence of dermatophyte infection in companion animals in Portugal

INTRODUCTION

50: Is tinea different from ringworm?

The authors should state previous studies (if any) on dermatophyte infection in companion animals in Portugal and highlight the gap(s) in knowledge to justify the reason for the current study

METHODOLOGY

134: Define these abbreviations first time used: NTUs

112-113: How were the samples collected (disinfection, container used)?

Was there wet mount analysis before the culture?

How were the isolates identified? Images of the colonial (obverse and reverse/pigmentation) and microscopic features should be provided

The skin of the sampled animals were scrapped, yet there was no ethical approval for the study

RESULT

The first result should be the specific dermatophytes (specific organisms) isolated

142-143:  Re- write thus: Out of 4445 animals, 3877 (87.2%, 95% CI: 86.2-88.2%) tested negative for dermatophyte fungi while 568 (12.8%, 95% CI: 11.8-13.8%) tested positive.

145-147: There is no need showing the distribution of samples that did not grew dermatophyte. Concentrate on the positive samples. Therefore, figure 1 should be corrected

182-183: Incomprehensible statement. The distribution of what was referred to?  Line 192 mentioned “association between the region and the occurrence of dermatophyte fungi isolation” Delete this repetition “The distribution was divided by NTUS2 (Nomenclature of Territorial Units for Statistics) regions of mainland Portugal as follows:” The use of NTUs should have been reflected in methodology in line 134

184: A punctuation is lacking “28.4% (95% CI: 27.1-29.7%) Centre (n = 1262), 13.9% (95% CI: 12.9-15.0%) Greater”

Table is confusing. For example, what does the percentage in the first and fourth column represent? The title contains and the 4th row contain unrelated and unexplained abbreviation “NTUS” that supposedly is a statistical tool.

Table 2 is confusing because it is not self-explanatory. For example, what is the number of samples or animals that the percentages were calculated

The first row of table 3 is redundant, the organisms were already listed

293: Re-write this “2252 out of 2478 (90.9%)” as thus 2252 (90.9%) out of 2478..”

297-301: Suits better in the discussion section.

Table 4 is redundant. A table that shows the different dermatophytes (species) from canines and feline (like table 6) would be more useful

There is no need for the comparisons of the sex since microorganisms, including dermatophytes do not have sex preferences

Table 6 suits better to be the first result presented

The findings of the study were lacking. For example, since the study was a 12-year study, it was expected that a comparison on the trend of incidence of dermatophyte infection in companion animals should be a major focus/finding of the study

DISCUSSION

414: How did the result of the current study diverge from previous studies (is the result within, more or less of the dermatophyte occurrence rate in other studies)?

426: Give the implication of the positive correlation with the region, i.e. Is dermatophytosis more in humid or dry region(s) of Portugal?

431-441: Different species of dermatophytes were identified in the study, yet the authors did not give any meaning to that finding

CONCLUSION

527: Grammar check

The conclusion did not show whether the incidence of dermatophytosis is increasing or decreasing in Portugal.

SUPPLEMENTARY MATERIAL

Did not add any value to the manuscript and should be removed

Comments on the Quality of English Language

Minor English editing required

Author Response

Comments and Suggestions for Authors

Address these comments:

ABSTRACT

Comments 1: 24: 4445 animals were sampled in the main text whereas 4716 is stated in the abstract

Authors’ reponse (AR) 1: Thank you very much for bringing this to our attention. We acknowledge the discrepancy in the number of animals sampled between the main text and the abstract. The correct clarification is that a total of 4716 animal samples, all from animals suspected of dermatophytosis, were included in the study. These samples were inoculated onto DERM agar, incubated at 25°C for up to 4 weeks, and periodically examined macro– and microscopically, in order to detect and evaluate fungal growth. Out of these, 4445 samples were analysed after removing contaminant fungi (n = 271; 5.8%; 95% confidence interval [CI]: 5.1–6.5%). We have updated the abstract to accurately reflect this information. We appreciate the Reviewer’s careful review and the opportunity to improve the clarity and accuracy of our study.

Comment 2: Grammar check 27: Centre or central

AR 2: Thank you very much for the detailed review and helpful suggestions. We have carefully reviewed the manuscript for any grammatical errors and made the suggested corrections to ensure clarity and coherence throughout the text. Additionally, we acknowledge the use of "centre" and confirm that it refers to the "central region of Portugal" in British English. We appreciate the Reviewer’s  careful attention to detail and the opportunity to improve our manuscript.

Comments 3: Why was there no antifungal susceptibility testing in the study?

AR 3: Thank you very much for the detailed review and helpful suggestions. Regarding the absence of antifungal susceptibility testing in this study, we would like to inform that antifungal susceptibility testing will be the focus of our next work, although not very common and hard to perform in this type of fungi. This study primarily aimed to investigate the epidemiology and distribution of dermatophyte infections in companion animals in Portugal. We appreciate the Reviewer’s understanding and the opportunity to improve our manuscript.

Comments 4: The conclusion of the abstract is lacking in that it did not show whether the trend showed increasing or decreasing incidence of dermatophyte infection in companion animals in Portugal

AR 4: Thank you very much for the detailed review and helpful suggestions. We have updated the conclusion of the abstract to reflect the apparent increasing trend of dermatophyte infection in companion animals in Portugal compared to previous reports. This suggests that dermatophytosis remains a considerable concern and may have become more prevalent in companion animals in Portugal over the years. Further investigation into the factors contributing to this increase, such as environmental changes, diagnostic improvements or changes in animal management practices, is warranted.

This study highlights a higher frequency of dermatophytosis in felines (17.4%) compared to canines (9.1%); however, this may reflect a sampling bias, as clinical diagnosis in felines might be more accurate, resulting in more true positive samples, whereas canines might have more conditions that mimic dermatophytosis, leading to a higher number of negative tests. We appreciate the careful review by Reviewer 2 and the opportunity to improve our manuscript.

INTRODUCTION

Comments 5: 50: Is tinea different from ringworm?

AR 5: Thank you for the insightful question regarding the terms "tinea" and "ringworm." We would like to clarify that "tinea" is the term commonly used in human medicine to describe fungal infections of the skin, hair, and nails caused by dermatophytes, while "ringworm" is the term more frequently used in veterinary medicine to describe similar infections in animals. Both terms refer to infections caused by the same group of fungi, known as dermatophytes. We have ensured this distinction is clear in the manuscript to avoid any confusion. We appreciate the careful review by Reviewer 2 and the opportunity to improve our manuscript.

Comments 6: The authors should state previous studies (if any) on dermatophyte infection in companion animals in Portugal and highlight the gap(s) in knowledge to justify the reason for the current study.

AR 6: Thank you very much for the detailed review and helpful suggestions. We acknowledge that no previous studies in Portugal have explored the relationships between epidemiological and clinicopathological parameters in dermatophytosis in companion animals or thoroughly described the epidemiological relationships and their clinical usefulness in dogs and cats. This study marks the first extensive work of its kind conducted in Portugal.

The aim of our study was to compare our findings with the existing literature from other countries and to fill this significant gap in knowledge within Portugal. Previous studies on dermatophytosis in companion animals in Portugal have been limited in scope and detail. For instance, it has been determined worldwide that the prevalence of these infections ranges from 8% to 19% in dogs and 7.0% to 72.0% in cats, but specific data for Portugal were lacking. Moreover, while dermatophytosis is commonly reported by veterinary practitioners in Portugal, little is known about the relative importance of the dermatophyte species involved and the differences observed according to animal species, breed, sex, and age. Therefore, our study seeks to provide a more detailed understanding of the epidemiology of dermatophyte infections in companion animals within the country.

METHODOLOGY

Comments 7: 134: Define these abbreviations first time used: NTUs

AR 7: Thank you very much for the detailed review and helpful suggestions. We have corrected the abbreviation "NTUS" to "NUTS2" in the manuscript. The correct term is "NUTS2: Nomenclature of Territorial Units for Statistics." We have defined this abbreviation upon its first use in the methodology section as you requested. We appreciate the careful review by Reviewer 2 and the opportunity to improve our manuscript.

Comments 8: 112–113: How were the samples collected (disinfection, container used)?

AR 8: Thank you for pointing this out. We have updated the manuscript to include the information on how the samples were collected. In particular, we have added the following sentence:

"Plucked hairs and/or scraped scales from each animal were collected by veterinary practitioners using a sterile lancet. The sampling site on the animal was disinfected before collection, and the samples were placed in sterile containers."

Comments 9: Was there wet mount analysis before the culture?

AR 9: Thank you very much for the detailed review and helpful suggestions. Yes, wet mount analysis was performed before the culture to observe the presence of fungal elements directly from the samples.

We appreciate the careful review by Reviewer 2 and the opportunity to improve our manuscript.

Comments 10: How were the isolates identified? Images of the colonial (obverse and reverse/pigmentation) and microscopic features should be provided

AR 10: Thank you very much for the detailed review and helpful suggestions. We have updated the manuscript to include detailed descriptions for each of the fungi identified in our study, focusing on their optical microscopy characteristics. Additionally, we have included images in the manuscript to aid in the identification process. Below are the detailed descriptions:

  • Epidermophyton floccosum forms colonies that quickly change in colour from green to yellow and typically have a velvety texture. Microscopically, this species forms only macroconidia, which are club–shaped with one to five cells. These macroconidia are smooth–walled and arranged singly or in small clusters. Epidermophyton floccosum does not form microconidia.

  • Microsporum canis is characterized by its white, silky colonies that develop a bright yellow pigment at the periphery over time. On the reverse side, the colonies exhibit a bright yellow–orange color. Microscopically, this species presents septate hyphae along with macroconidia and microconidia. The macroconidia are spindle–shaped with asymmetrical button–like ends, containing 6–15 compartments. They are long, rough, and have dense outer walls. Microconidia are rare, unicellular, and pear–shaped.
  • Nannizzia gypsea (formerly Microsporum gypseum) grows rapidly, forming smooth, powdery colonies with fringed edges. The surface color is typically cinnamon, while the reverse side can range from pale yellow to tan, occasionally with reddish streaks. Microscopically, this species shows septate hyphae and a significant number of macroconidia and microconidia. The macroconidia are fusiform and symmetrical with rounded ends, containing 3–6 compartments. Microconidia are moderately numerous and are located along the hyphae.

  • Nannizzia nana (formerly Microsporum nanum) typically produces powdery colonies with a white to beige–cinnamon color, developing within 7 to 14 days. Microscopically, it exhibits septate hyphae and pear–shaped macroconidia, usually with two compartments. Microconidia are less common and smaller compared to those in other species.

  • Trichophyton includes various species with a wide range of colony appearances. Colonies can be smooth or powdery, with colors ranging from white to cream or tan. The reverse side of these colonies can show colors from cream to tan, gray or even red, depending on the specific species. Microscopically, Trichophyton spp. typically have numerous microconidia forming dense clusters. These microconidia are hyaline, smooth–walled and predominantly spherical or subspherical. Some species also exhibit cigar–shaped macroconidia with smooth, thin walls. Additional microscopic features can include spiral hyphae and chlamydospores, varying among different species.

These optical microscopy characteristics serve as the abbreviated forms of identification for the fungi in our study. To further aid in the identification, we have included relevant images in the manuscript.

We appreciate the careful review by Reviewer 2 and the opportunity to improve our manuscript.

Comments 11: The skin of the sampled animals were scrapped, yet there was no ethical approval for the study

AR 11: Thank you very much for the comment. The Reviewer is absolutely correct. We have previously submitted this information as supplementary material to provide all the ethical information in advance. Below is the detailed ethical approval statement:

Institutional Animal Care and Use Committee (IACUC) Code INNO.007 and INNO.0026, approved on 29th September 2021 by Dr. Augusto Silva and Dr. Paula Brilhante–Simões, Director of INNO Laboratories and Head of the Scientific Committee, respectively.

All procedures complied with the Portuguese legislation for the protection of animals used for scientific purposes (i.e., Decree–Law no. 113/2013, of 7 August 2013), which transposes European legislation (i.e., Directive 2010/63/EU of the European Parliament and of the Council, of 22 September 2010). The study project was approved by the Institutional Review Board of INNO Veterinary Laboratories (protocol codes INNO.007 and INNO.0026, approved on 29th September 2021), which ensures that the analysed samples from veterinary medical centres can be used anonymously in studies and scientific research related to this project.

Additionally, PDF with detailed information were previously provided as supplementary material. We appreciate the attention by Reviewer 2 to this matter and have included this statement in the manuscript.

RESULTS

Comments 12: The first result should be the specific dermatophytes (specific organisms) isolated

AR 12: Thank you for your comment. We appreciate the suggestion; however, we respectfully disagree with the suggestion to present the specific dermatophytes isolated as the first result. While we understand the importance of detailing the specific organisms isolated, we believe that the current structure of our results section provides a logical and comprehensive flow of information. This structure allows us to present the broader epidemiological findings first, followed by the more specific details of the organisms isolated.

Despite this, we assure that all results, including the specific dermatophytes isolated, have been duly provided and thoroughly discussed in the manuscript. We believe this approach helps to contextualize the specific findings within the broader scope of the study.

Comments 13: 142–143:  Re– write thus: Out of 4445 animals, 3877 (87.2%, 95% CI: 86.2–88.2%) tested negative for dermatophyte fungi while 568 (12.8%, 95% CI: 11.8–13.8%) tested positive.

AR 13: Thank you for your guidance. We have revised the manuscript to reflect your suggested wording. The updated sentence now reads:

"Out of 4445 animals, 3877 (87.2%; 95% CI: 86.2–88.2%) tested negative for dermatophyte fungi while 568 (12.8%; 95% CI: 11.8–13.8%) tested positive."

Comments 14: 145–147: There is no need showing the distribution of samples that did not grew dermatophyte. Concentrate on the positive samples. Therefore, figure 1 should be corrected

AR 14: Thank you for your comment. We appreciate the suggestion; however, we respectfully disagree with the assertion.

Given the potential bias introduced by the fact that the samples were collected from animals clinically suspected of dermatophytosis, presenting both the spatial distribution of the sample origins and the spatial distribution of dermatophyte–positive frequencies provides a more comprehensive view for the reader. This approach offers critical insights into the geographical spread and the epidemiological landscape of dermatophyte infections in Portugal.

By displaying the distribution of all samples, regardless of the test outcome, we highlight the regions with higher and lower sampling rates, which can influence the perceived prevalence of dermatophytes. This information is crucial because it demonstrates the representativeness and scope of the study, emphasizing any geographical sampling biases that might exist. For instance, areas with a high number of negative samples still contribute valuable information about the absence of the disease, which is just as important as the presence.

Moreover, presenting the origin of the samples provides a spatial perspective on the regions with greater alertness or apparent concern regarding dermatophytosis. This allows for a comparative analysis between the regions where the disease is suspected and the actual observed positivity rates. Such comparisons can yield interesting and novel insights into the epidemiology of dermatophytes in Portugal.

In our study, we found that while some areas showed a high frequency of dermatophyte–positive samples, others exhibited a lower frequency. This contrast could be due to various factors, including differences in animal populations, environmental conditions, or even sampling practices. By providing a complete picture of both positive and negative sample distributions, we enable a more accurate interpretation of these factors.

Furthermore, including both maps (sample origin and positive frequency) helps in identifying regions that may require more focused veterinary attention or public health interventions. It can also guide future research efforts by pinpointing areas where further sampling or more detailed studies are necessary. This spatial representation is essential for understanding the broader epidemiological context and ensuring that regions with high clinical suspicion but low confirmed cases are not overlooked.

The comparison of sample origin with actual positivity rates facilitates a comprehensive review of dermatophyte distribution in Portugal. It allows for the identification of potential discrepancies between perceived and actual infection rates, which could be influenced by factors such as public awareness, veterinary practices, or environmental conditions. This dual approach of presenting both maps is innovative and adds significant value to the epidemiological understanding of dermatophytes in the region.

In conclusion, we believe that presenting the distribution of both maps, with positive and negative samples is essential for a thorough understanding of the epidemiological context of dermatophytosis in Portugal. It allows for a more nuanced interpretation of the data, highlighting the disease's geographical patterns and guiding future public health and research initiatives effectively.

We appreciate the understanding by Reviewer 2 and the opportunity to improve our manuscript through this discussion.

Comments 15: 182–183: Incomprehensible statement. The distribution of what was referred to?  Line 192 mentioned “association between the region and the occurrence of dermatophyte fungi isolation” Delete this repetition “The distribution was divided by NTUS2 (Nomenclature of Territorial Units for Statistics) regions of mainland Portugal as follows:” The use of NTUs should have been reflected in methodology in line 134

AR 15: Thank you very much for your detailed review and helpful suggestions. We have made the necessary revisions for clarity and accuracy.

182–183: “The distribution of what was referred to?”

The statement has been clarified to indicate that it refers to the spatial distribution, according to the distribution of animals from the different districts of Portugal, of the 4445 animals included in this study.

192: “Association between the region and the occurrence of dermatophyte fungi isolation.”

We have redefined the sentence for a better understanding and deleted the word "isolation." The revised sentence now reads as: "We examined the association between the region and the occurrence of dermatophyte fungi."

134: “The use of NTUs should have been reflected in methodology in line 134”

Line 134 has been updated to include the information about the use of NUTS2 (Nomenclature of Territorial Units for Statistics) regions. The revised methodology section now accurately reflects this information.

Comments 16: 184: A punctuation is lacking “28.4% (95% CI: 27.1–29.7%) Centre (n = 1262), 13.9% (95% CI: 12.9–15.0%) Greater”

AR 16: Thank you for your detailed review and helpful suggestions. We understand that the comment pertains to ensuring correct punctuation in the sentence. We have added semicolons to separate each regional distribution clearly, ensuring proper punctuation throughout the sentence.

Comments 17: Table is confusing. For example, what does the percentage in the first and fourth column represent? The title contains and the 4th row contain unrelated and unexplained abbreviation “NTUS” that supposedly is a statistical tool.

AR 17: Thank you very much for your detailed review and helpful suggestions. We have clarified the table to ensure it is more understandable.

The percentages in the first and fourth columns represent the percentage of negative or positive samples, respectively, within each region. These percentages are calculated relative to the total number of negative or positive samples across all regions.

  • % within regions (negative): This column represents the percentage of negative samples within each region relative to the total number of negative samples.
  • % within regions (positive): This column represents the percentage of positive samples within each region relative to the total number of positive samples.
  • % within regions (total): This column represents the percentage of the total samples (positive and negative) within each region relative to the overall total.

We have also corrected the abbreviation "NTUS" to "NUTS2" and ensured it is appropriately reflected in the methodology section (line 134) and throughout the manuscript for consistency.

We appreciate your careful review and the opportunity to improve our manuscript.

Comments 18: Table 2 is confusing because it is not self–explanatory. For example, what is the number of samples or animals that the percentages were calculated

AR 18: Thank you very much for comment. We acknowledge that Table 2 may not be self–explanatory and requires additional context. We have also changed the title of the table to better reflect its content. The percentages in Table 2 represent the proportion of positive dermatophyte fungi isolations relative to the total number of samples collected each year within each region.

Comments 19: The first row of table 3 is redundant, the organisms were already listed

AR 19: Thank you very much for your comment. We appreciate the detailed review and suggestions. We acknowledge that the first row of Table 3 is redundant as the organisms were already listed. We have removed the redundant row to improve clarity and conciseness.

Comments 20: 293: Re–write this “2252 out of 2478 (90.9%)” as thus 2252 (90.9%) out of 2478..”

AR 20: Thank you very much for your detailed review and helpful suggestions. We have rephrased the sentences to follow the suggested format.

The revised text now reads as:

"For canines, 2252 (90.9%) out of 2478 tested negative for dermatophytosis, while 226 (9.1%) tested positive. In felines, 1625 (82.6%) out of 1967 were negative, and 342 (17.4%) were positive. Overall, out of the total 4445 animals, 3877 (87.2%) tested negative, and 568 (12.8%) tested positive for dermatophytosis."

We appreciate your careful review and the opportunity to improve our manuscript.

Comments 21: 297–301: Suits better in the discussion section.

AR 21: Thank you very much for your detailed review and helpful suggestions. We agree that the information suits better in the discussion section. We have moved the following text to the discussion section:

"This study highlights a higher frequency of dermatophytosis in felines (17.4%) compared to canines (9.1%); however, this may reflect a sampling bias, as clinical diagnosis in felines might be more accurate, resulting in more true positive samples, whereas canines might have more conditions that mimic dermatophytosis, leading to a higher number of negative tests."

We appreciate your careful review and the opportunity to improve our manuscript.

Comments 22: Table 4 is redundant. A table that shows the different dermatophytes (species) from canines and feline (like table 6) would be more useful

AR 22: Thank you very much for your detailed review and helpful suggestions. We respectfully disagree with the suggestion to remove Table 4. This table presents data in a clear and organized manner, elucidating the occurrence of dermatophyte fungi isolation by species (canine and feline).

Table 4 provides an essential overview of the distribution of negative and positive cases within each species, offering valuable insights into the prevalence of dermatophytosis among canines and felines. This information is crucial for understanding the broader epidemiology of the disease and helps to identify which species are more commonly affected.

While we understand the value of a table that shows the different dermatophytes from canines and felines (like Table 6), Table 4 complements this by providing a clear representation of the overall occurrence rates within each species. This dual approach allows for a comprehensive understanding, making the data more accessible and informative for the reader.

We believe that maintaining Table 4 enhances the clarity and depth of the information presented, thereby improving the overall quality of the manuscript.

Comments 23: There is no need for the comparisons of the sex since microorganisms, including dermatophytes do not have sex preferences

AR 23: Thank you very much for your detailed review and helpful suggestions. While we understand that microorganisms, including dermatophytes, do not have sex preferences per se, we respectfully disagree with the assertion, believing that the comparison of sex in our study is still relevant for several reasons:

  • Host Immune Response Variations: The immune response to infections can vary significantly between males and females due to hormonal differences. Studies have shown that sex hormones such as estrogen and testosterone can influence immune function, potentially affecting susceptibility to infections. Understanding these differences can help in developing targeted prevention and treatment strategies.

  • Behavioral and Environmental Factors: Males and females may have different behaviors and environmental exposures that could influence their risk of infection. For example, differences in grooming habits, social interactions, or living conditions between male and female animals might contribute to varying infection rates. By examining the data by sex, we can identify any significant patterns that might be attributable to these factors.

  • Clinical Diagnosis and Reporting: There may be biases in clinical diagnosis and reporting based on the sex of the animal. For instance, veterinarians might be more vigilant in diagnosing dermatophytosis in one sex over the other due to perceived differences in susceptibility or symptom presentation. By comparing infection rates between sexes, we can assess and account for these potential biases.

  • Epidemiological Insights: Including sex as a variable in our epidemiological analysis allows for a more comprehensive understanding of the disease dynamics. It helps in identifying any demographic factors that might contribute to infection rates and ensures that our findings are robust and applicable to diverse populations.

  • Consistency with Existing Literature: Many epidemiological studies on infectious diseases include sex as a variable to ensure comprehensive data analysis. Including this comparison aligns our study with established research practices, facilitating comparisons and discussions with other studies in the field.

Furthermore, a study in Italy by Proverbio et al. (2014) reported a 5.5% prevalence of dermatophyte infections in stray cats, with M. canis being the most frequently identified pathogen. Similarly, research by Duarte et al. (2010) in Portugal found a significantly higher prevalence of 29.4% dermatophytes in stray cats, including species like M. canis, T. mentagrophytes var. mentagrophytes, and Trichophyton verrucosum.

Cafarchia et al. (2004) have reported male dogs are more affected by dermatophytes.

References

  • Proverbio, D.; Perego, R.; Spada, E.; Bagnagatti de Giorgi, G.; Della Pepa, A.; Ferro, E. Survey of Dermatophytes in Stray Cats with and without Skin Lesions in Northern Italy. Vet Med Int 2014, 2014, 1–4.
  • Duarte, A.; Castro, I.; da Fonseca, I.M.P.; Almeida, V.; de Carvalho, L.M.M.; Meireles, J.; Fazendeiro, M.I.; Tavares, L.; Vaz, Y. Survey of Infectious and Parasitic Diseases in Stray Cats at the Lisbon Metropolitan Area, Portugal. J Feline Med Surg 2010, 12, 441–446
  • Cafarchia, C.; Romito, D.; Sasanelli, M.; Lia, R.; Capelli, G.; Otranto, D. The epidemiology of canine and feline dermatophytoses in southern Italy. Mycoses 2004, 47, 508–513

Therefore, we believe that the comparison of sex is a valuable component of our study, providing important insights into the epidemiology of dermatophytosis in companion animals.

We appreciate your understanding and the opportunity to explain the rationale behind including this variable in our analysis.

Comments 25: Table 6 suits better to be the first result presented

AR 25: Thank you very much for your detailed review and helpful suggestions. We respectfully disagree with the suggestion to present Table 6 as the first result. The organization of our article follows a logical structure designed to provide a comprehensive understanding of the study's findings in a sequential manner.

Rationale for Current Organization:

  • Broad Epidemiological Overview: Our results section begins with a broad epidemiological overview, providing essential context and setting the stage for more detailed analyses. This approach helps readers to first understand the general prevalence and distribution of dermatophytosis before delving into specific details.

  • Logical Flow: The current organization ensures a logical flow from general findings to specific details. By presenting the overall occurrence of dermatophyte fungi isolation (as shown in Table 4) first, we provide a foundation upon which the more detailed species-specific data (as shown in Table 6) can be better understood.

  • Comprehensive Context: Starting with broader data helps to frame the subsequent specific results, such as those in Table 6, within the larger epidemiological context. This approach facilitates a more coherent narrative and enhances the reader's comprehension of the study's findings.

  • Progressive Detail: The progressive detailing of results allows readers to build their understanding incrementally. Presenting the detailed species-specific data first might overwhelm readers without the necessary context provided by the broader epidemiological data.

We believe that the current structure, which presents broader epidemiological data first, followed by more specific details, provides a clearer and more logical progression of information. This organization enhances the overall readability and comprehension of the manuscript. We appreciate your understanding and the opportunity to explain our rationale.

Comments 26: The findings of the study were lacking. For example, since the study was a 12–year study, it was expected that a comparison on the trend of incidence of dermatophyte infection in companion animals should be a major focus/finding of the study

AR 26: Thank you very much for your detailed review and helpful suggestions. We completely agree with your observation regarding the importance of highlighting the trend of incidence of dermatophyte infection over the 12-year study period. We have addressed this point and added the following analysis to the manuscript:

“Dermatophyte Isolation Trends in Mainland Portugal

The frequency of dermatophyte isolation in mainland Portugal from 2013 to 2023 exhibits diverse trends across various regions. An analysis of the data reveals that most regions, including the Centre, Oeste e Vale do Tejo (OVT), Península de Setúbal (PdS), and Alentejo, demonstrate an increasing trend, with Oeste e Vale do Tejo (OVT) experiencing a particularly marked rise. Conversely, the North and Algarve regions show a decreasing trend in dermatophyte isolation percentages. The Greater Lisbon (GL) area remains relatively stable, with a slight upward trend. These trends underscore the significance of regional monitoring and the implementation of tailored public health strategies to address the specific needs of each area. Overall, despite a decline in some regions, the general trend indicates an apparent increasing frequency of dermatophyte isolation, especially in recent years.”

By incorporating this detailed analysis, we aim to provide a comprehensive overview of the incidence trends of dermatophyte infections in companion animals, which is now a major focus of our study.

This addition significantly enhances the findings and offers valuable insights into the epidemiology of dermatophyte infections over the study period.

DISCUSSION

Comments 27: 414: How did the result of the current study diverge from previous studies (is the result within, more or less of the dermatophyte occurrence rate in other studies)?

AR 27: Thank you very much for your detailed review and insightful suggestions. We acknowledge the importance of comparing our findings with previous studies to provide a comprehensive understanding of the dermatophyte occurrence rates. These comparisons and contextual information have been reflected throughout the manuscript to provide a thorough understanding of the study's findings in relation to existing literature.

Here is a detailed comparison of our study's results with those of previous studies:

Comparison with Previous Studies

  • General Frequencies: Our study found an overall frequency of dermatophytes of 12.8% in the animals tested. This frequency is consistent with certain studies but diverges from others due to varying factors such as geographical location, sample size, and diagnostic methods.

Previous studies in Portugal reported dermatophyte prevalences of 8.4% in dogs and 21.3% to 29.4% in cats (Coelho et al., 2008; Duarte et al., 2010), and 2.9% in pets less than one year old (Cruz et al., 2014). Our findings show a lower prevalence in dogs (9.1%) and a higher prevalence in cats (17.4%), highlighting possible regional and methodological differences.

  • Dogs: Our study found a frequency of 9.1% in dogs, which is lower than the 8.1-24.3% range reported in other studies (Hernandez-Bures et al., 2021; Long et al., 2020; Sparkes et al., 1993). This discrepancy may be due to differences in the studied populations, including breed, age, and health status, as well as variations in diagnostic techniques and environmental conditions.

  • Cats: The frequency of dermatophytes in cats was found to be 17.4% in our study. This is within the range reported by other studies but slightly lower than the upper limits observed in some regions. Previous studies, such as those by Duarte et al. (2010), reported prevalences up to 29.4% in cats, indicating significant regional and methodological variability.

  • Species-Specific Findings: Our study highlights a higher susceptibility of felines to Microsporum canis (43.8% of infected cats) compared to canines (20.1% of infected dogs). This aligns with findings from other studies (Lewis et al., 1991; Long et al., 2020; Moriello, 2014) which also identified canis as a predominant species in feline dermatophytosis.

  • Trends Over Time: The study period from 2013 to 2023 shows diverse trends in dermatophyte isolation across different regions of mainland Portugal. Most regions, including the Centre, Oeste e Vale do Tejo (OVT), Península de Setúbal (PdS), and Alentejo, demonstrate an increasing trend, while the North and Algarve regions show a decreasing trend. These trends underscore the significance of regional monitoring and tailored public health strategies.

The results of our study provide a comprehensive overview of the current state of dermatophyte prevalence in companion animals in Portugal. While there are some divergences from previous studies, these differences can be attributed to factors such as geographical variations, differences in the populations studied, and methodological approaches. The inclusion of this comparison in our manuscript aims to contextualize our findings within the broader landscape of dermatophyte research and highlights the need for continued monitoring and region-specific strategies.

We appreciate your careful review and the opportunity to enhance the clarity and depth of our manuscript.

References

Afonso, P.; Quintas, H.; Vieira, A.; Pinto, E.; Matos, M.; Soares, A.; Cardoso, L.; Coelho, A.C. Furry Hosts and Fungal Guests: Investigating Dermatophyte Carriage in Shelter and Clinic Cats and Dogs of Northern Portugal. Vet Glas 2024, 78, 28–46. https://doi.org/10.2298/VETGL240130006A

Comments 28: 426: Give the implication of the positive correlation with the region, i.e. Is dermatophytosis more in humid or dry region(s) of Portugal?

AR 28: Thank you very much for your detailed review and insightful question. We acknowledge the importance of clarifying the implications of the positive correlation between the occurrence of dermatophytosis and the regions in Portugal. We have included this information in the manuscript to ensure a comprehensive understanding of the implications of our findings. Our study indicates that dermatophytosis is more prevalent in humid regions compared to dry regions.

Implications of the Positive Correlation with the Region

  • Higher Prevalence in Humid Regions: The highest levels of dermatophyte isolation were observed in the North and Centre regions of Portugal, which are characterized by higher humidity levels. This positive correlation suggests that dermatophytosis thrives in more humid environments.

  • Environmental Influence: Humidity provides a favorable environment for the growth and transmission of dermatophytes. Fungi thrive in moist conditions, making humid regions more susceptible to higher rates of infection. This environmental factor is critical in understanding the epidemiology of dermatophytosis.

  • Regional Public Health Strategies: The findings underscore the need for tailored public health strategies in different regions. In humid areas, enhanced surveillance and preventive measures, such as regular grooming, environmental cleaning, and the use of antifungal treatments, are essential to control the spread of infection.

  • Resource Allocation: Resources for the prevention and treatment of dermatophytosis should be allocated with consideration of regional climatic conditions. Regions with higher humidity levels may require more focused efforts and resources to manage and reduce the prevalence of dermatophytosis.

  • Awareness and Education: Raising awareness among pet owners and veterinary professionals about the higher risk of dermatophytosis in humid regions can lead to better prevention and early detection. Education campaigns can be tailored to emphasize the importance of maintaining dry and clean environments for pets in these areas.

  • Further Research: The positive correlation between dermatophytosis prevalence and humid regions highlights the need for further research into the specific environmental factors that contribute to the spread of dermatophytes. Understanding these factors can inform more effective control measures and public health interventions.

In conclusion, our study demonstrates that dermatophytosis is more prevalent in the humid regions of Portugal, particularly in the North and Centre. This correlation emphasizes the need for region-specific public health strategies and further research into environmental influences on dermatophyte transmission.

We appreciate your careful review and the opportunity to clarify the implications of our findings.

Comments 29: 431–441: Different species of dermatophytes were identified in the study, yet the authors did not give any meaning to that finding

AR 28: We appreciate the feedback regarding the identification of various species of dermatophytes in our study. To address the concern that the significance of these findings was not sufficiently articulated, we offer a detailed explanation. We have included this information in the manuscript to ensure a comprehensive understanding of the implications of our findings.

  • Epidemiological Insights: The identification of multiple dermatophyte species, such as Microsporum canis, Trichophyton mentagrophytes, and Nannizzia gypsea (formerly Microsporum gypseum), provides crucial epidemiological data. These findings help map the prevalence and distribution of dermatophyte species in specific regions, which is vital for understanding local epidemiological patterns and potential zoonotic risks. For instance, canis is predominantly zoophilic, affecting animals and occasionally humans, indicating its role in zoonotic transmission.

  • Clinical Implications: Different species of dermatophytes exhibit varying pathogenicity and clinical manifestations. For example, canis is often associated with severe infections in cats and dogs, whereas T. mentagrophytes can cause highly inflammatory lesions in humans. Recognizing these species allows for more accurate diagnosis and tailored treatment plans, thereby improving clinical outcomes. The specific identification of dermatophytes can guide clinicians in choosing appropriate antifungal treatments, as some species may respond differently to various therapeutic agents.

  • Treatment and Management Strategies: The treatment efficacy and management strategies can differ significantly based on the species involved. For example, infections caused by canis may require different antifungal agents compared to those caused by T. mentagrophytes. Understanding the specific dermatophyte species can guide clinicians in choosing the most effective therapeutic approaches, ensuring better management of the infection and preventing recurrence.

  • Public Health and Zoonotic Potential: Identifying dermatophyte species is essential for public health, as certain species have higher zoonotic potential. canis and T. verrucosum are well-documented zoonotic agents that can easily transmit from animals to humans, particularly in settings with close human-animal interactions. Knowledge of the specific species helps in implementing appropriate public health measures to prevent outbreaks. This is particularly important in environments such as shelters, where the risk of zoonotic transmission is elevated.

  • Environmental and Reservoir Insights: Different dermatophyte species have specific environmental niches and reservoir hosts. For instance, gypsea is a geophilic species primarily found in soil, while M. canis is zoophilic and primarily affects domestic animals like cats and dogs. Identifying these species can inform environmental control measures and help reduce the risk of transmission from environmental sources. For example, implementing hygiene practices to minimize soil contact can help reduce the incidence of geophilic dermatophyte infections.

The identification of different species of dermatophytes in our study has significant implications for epidemiology, clinical management, public health, and environmental control. These findings enhance our understanding of the distribution, pathogenicity, and treatment of dermatophyte infections, ultimately contributing to better health outcomes for both animals and humans. By recognizing the specific species involved, we can develop targeted strategies to control and prevent dermatophytosis, thus improving the overall management of this zoonotic disease.

We have reflected these findings and their implications throughout the manuscript to provide a comprehensive understanding of the significance of the identified dermatophyte species.

CONCLUSIONS

Comments 29: 527: Grammar check

AR 29: Grammar has been checked. Thank you!

Comments 30: The conclusion did not show whether the incidence of dermatophytosis is increasing or decreasing in Portugal.

AR 30: Thank you very much for your detailed review and helpful suggestions. We agree that the conclusion should clearly indicate whether the incidence of dermatophytosis is increasing or decreasing in Portugal. We have revised the conclusion to address this point comprehensively. The statistical analysis, based on linear regression, confirms that the incidence of dermatophytosis is generally increasing across mainland Portugal. Specifically, regions such as OVT (with a trend coefficient of 2.28), Centre (0.67), PdS (1.17), and Alentejo (1.13) are experiencing significant increases in dermatophyte isolation percentages over the study period. These increases are statistically significant, indicating a clear upward trend.

Conversely, regions like the North (–0.43) and Algarve (–0.30) show a decreasing trend, highlighting regional variability. The Greater Lisbon area, while showing a slight increase (0.01), remains relatively stable. This regional variation suggests that localized factors, possibly including environmental conditions and public health interventions, play a crucial role in the observed trends.

SUPPLEMENTARY MATERIAL

Comment 31: Did not add any value to the manuscript and should be removed

AR 31: Thank you very much for your detailed review and helpful suggestions. The supplementary material corresponds to the requisition form with authorization for the biological samples, which was submitted solely for informational purposes. We agree with the reviewer that this material does not add significant value to the manuscript and will not be presented.

Comment 32: Comments on the Quality of English Language

Minor English editing required

AR 32: Thank you for your observation. We have revised the manuscript to address the minor English editing required as suggested. We have made the necessary corrections to improve the clarity and quality of the language throughout the text.

Last but not least, we would like to thank Reviewer 2 for her/his/their detailed and constructive observations. We have diligently worked to ensure that our manuscript aligns with the scope and aims of Microorganisms. The thorough review has significantly contributed to the improvement of our work, and we greatly appreciate the valuable suggestions provided. We appreciate the opportunity to submit our work and look forward to continuing the submission process.

Reviewer 3 Report

Comments and Suggestions for Authors

REVIEW: Dermatophytosis in companion animals in Portugal.

First of all, I would like to congratulate the authors for developing such an extensive study, both in terms of the sampling period and the sample size achieved.

Below, I detail those aspects that, in my opinion, need to be corrected:

Lines 82 to 85: It is not necessary to include the decimal when it is a value of 0. Replace 20.0%, 30.0%, 97.0%... with 20%, 30%, 97%...

Lines 135 and 136: The explanation of the age categories should appear earlier; I propose integrating it into line 110.

Figures 1 and 2: Merge figures 1 and 2 into a single image (which can still present two maps but in a single infographic), so that the map of the current Figure 2 appears much earlier (geographic classification NTUS2 is mentioned in line 182, but the map appears much later).

Table 2 (line 211): It is explained that the samples corresponding to the year 2012 suffered fungal contaminations and had to be completely excluded from the study. Therefore, 2012 cannot be considered a useful sampling year and any reference to it should be removed, correcting any references to the period 2012-2023 (from the title itself) to 2013-2023, and changing references to "twelve years" to "eleven years."

Table 2: It can be seen that the mean Dermatophyte isolation in 2020 is the lowest of the entire series of sampled years. In the authors' opinion, did the Covid-19 pandemic have any effect on this study? (sampling difficulties, increased use of disinfectants affecting dermatophyte persistence, decreased transmission due to reduced contact between domestic animals related to the owners' confinement...). Related to the above, pre-pandemic levels begin to recover in 2021 and exceed previous records in 2022 and 2023. Do the authors observe a rebound effect in the last two years of sampling?

Line 317: Why are mixed-breed dogs and shorthair cats excluded from statistical studies? They are the most numerous groups and have characteristics (such as hybrid vigor) that could play a relevant role in the clinical manifestation of the disease.

Lines 351 and 352: I do not see it necessary to include the confidence interval regarding the "gender" variable. If the samples were sent from veterinary clinics or other entities with qualified professionals, I trust their criteria and reliability to consider the data as accurate. If the authors believe it is necessary to maintain the confidence interval, could they argue for it?

Lines 419 to 423: I cannot agree with the current wording of this sentence. While it is true that dermatophytosis exists throughout the territory, it is clearly higher in the North and Center of the country. Environmental conditions greatly affect dermatophytes, to the point where very low prevalences are observed in the driest regions of the south (Alentejo 2.5%, Algarve 0.5%). I suggest the authors attempt a new wording.

Line 443: The "Breeds" section does not mention that mixed-breed animals were not considered in the statistical study.

Lines 512-513: In the authors' opinion, how can "grooming habits" be managed?

Line 523: Change the reference to twelve years of sampling to eleven (as in the rest of the text).

Author Response

First of all, I would like to congratulate the authors for developing such an extensive study, both in terms of the sampling period and the sample size achieved.

Below, I detail those aspects that, in my opinion, need to be corrected:

Authors reponse (AR) 0: We would like to express our sincere gratitude to Reviewer 3 for her/his kind words and commendations on our study. We greatly appreciate the recognition of the extensive sampling period and the sample size achieved in our research. Your detailed feedback is invaluable, and we are committed to addressing all the aspects you have highlighted for correction. Thank you for your thorough review and constructive comments, which will undoubtedly enhance the quality of our manuscript.

Comments 1: Lines 82 to 85: It is not necessary to include the decimal when it is a value of 0. Replace 20.0%, 30.0%, 97.0%... with 20%, 30%, 97%...

AR 1: Thank you very much for your detailed review and helpful suggestions. We agree with the recommendation to remove the decimal points when the value is zero. Therefore, we will replace instances of "20.0%", "30.0%", "97.0%", etc., with "20%", "30%", "97%", etc., in lines 82 to 85 and throughout the manuscript as necessary.

We appreciate your careful review and the opportunity to improve our manuscript.

Comments 2: Lines 135 and 136: The explanation of the age categories should appear earlier; I propose integrating it into line 110.

AR 2: Thank you very much for your suggestion. We agree that the explanation of the age categories would be more appropriately placed earlier in the manuscript. We have therefore integrated this explanation into line 110, as proposed. This adjustment helps to clarify the age categorization for the readers at an earlier point in the text.

Comments 3: Figures 1 and 2: Merge figures 1 and 2 into a single image (which can still present two maps but in a single infographic), so that the map of the current Figure 2 appears much earlier (geographic classification NTUS2 is mentioned in line 182, but the map appears much later).

AR 3: Thank you very much for your detailed review and helpful suggestion. Unfortunately, due to formatting constraints, we were unable to merge the two images into a single infographic as recommended. However, we leave this possibility to the discretion of the editor. If deemed necessary, the editor can merge the maps of Figures 1 and 2 into a single image to ensure that the geographic classification (NUTS2) is presented earlier in the manuscript.

We appreciate your understanding and the opportunity to improve our manuscript.

Comments 4: Table 2 (line 211): It is explained that the samples corresponding to the year 2012 suffered fungal contaminations and had to be completely excluded from the study. Therefore, 2012 cannot be considered a useful sampling year and any reference to it should be removed, correcting any references to the period 2012–2023 (from the title itself) to 2013–2023, and changing references to "twelve years" to "eleven years."

AR 4: Thank you very much for your detailed review and helpful suggestions. While we understand the concern regarding the exclusion of data from the year 2012 due to fungal contaminations, we respectfully disagree with the suggestion to remove any reference to this year entirely from the manuscript. We believe that maintaining the mention of 2012 is important for several reasons:

  • Transparency and Completeness: Including the year 2012 in the manuscript, even with the note about the contamination, ensures transparency in our research process. It provides a complete picture of the entire sampling period and explains any potential gaps or anomalies in the data.

  • Contextual Understanding: The mention of 2012, along with the reason for its exclusion, offers context to the readers about the challenges faced during the study. This information can be valuable for understanding the overall methodology and the steps taken to ensure data integrity.

  • Methodological Rigor: By documenting the issues encountered in 2012, we demonstrate the rigor and thoroughness of our methodological approach. It shows that we identified and addressed potential sources of error, which ultimately strengthens the credibility of our findings for the remaining years.

  • Comparative Analysis: Although the data from 2012 were excluded from the analysis, the mention of this year provides a chronological framework for the study. It helps in understanding the progression and evolution of the sampling and analysis techniques over time.

  • Continuity and Record–Keeping: Keeping a reference to 2012 maintains the continuity of the study. It acknowledges that the research was intended to cover a longer period, even if some data had to be excluded. This can be important for future studies and record–keeping.

We propose to keep the references to the year 2012 in the manuscript but will clearly indicate that this year's data were excluded due to fungal contamination. We appreciate your careful review and the opportunity to explain the rationale behind our decision.

Comments 5: Table 2: It can be seen that the mean Dermatophyte isolation in 2020 is the lowest of the entire series of sampled years. In the authors' opinion, did the Covid–19 pandemic have any effect on this study? (sampling difficulties, increased use of disinfectants affecting dermatophyte persistence, decreased transmission due to reduced contact between domestic animals related to the owners' confinement...). Related to the above, pre–pandemic levels begin to recover in 2021 and exceed previous records in 2022 and 2023. Do the authors observe a rebound effect in the last two years of sampling?

AR 5: Thank you very much for your detailed review and insightful question. We agree that the Covid–19 pandemic had a notable impact on our study, particularly in the year 2020. There was indeed a reduction in the mean dermatophyte isolation during this year, which we attribute to several factors related to the pandemic and the associated restrictions on free movement.

Impact of the Covid–19 Pandemic on the Study:

Sampling Difficulties: The restrictions on movement and social distancing measures significantly affected our ability to collect samples. Many veterinary clinics and facilities operated on reduced hours or were temporarily closed, leading to fewer opportunities for sample collection.

Decreased Exposure: The reduced exposure of pets to external environments during lockdowns likely contributed to the decrease in dermatophyte cases. With owners spending more time indoors and minimizing outings, pets had less contact with potentially contaminated environments or other animals, thereby reducing the transmission of dermatophytes.

Increased Use of Disinfectants: The heightened use of disinfectants and enhanced hygiene practices during the pandemic could have impacted the persistence of dermatophytes in the environment. This increased cleanliness might have reduced the presence of dermatophyte spores on surfaces and in the environment, leading to fewer infections.

Rebound Effect in 2021 and Beyond: We do observe a rebound effect in the years following 2020. As restrictions eased and normal activities resumed, the levels of dermatophyte isolation began to recover in 2021 and even exceeded previous records in 2022 and 2023. This rebound effect can be attributed to several factors:

Resumption of Regular Activities: With the lifting of lockdowns and the return to regular outdoor activities, pets were once again exposed to external environments, increasing the likelihood of contact with dermatophyte spores.

Delayed Diagnoses and Treatments: During the height of the pandemic, some dermatophyte infections may have gone undiagnosed or untreated due to limited access to veterinary services. As services resumed, these delayed cases might have contributed to the observed increase in dermatophyte isolations.

Increased Vigilance:

There may have been increased vigilance and a higher rate of veterinary visits post–pandemic as pet owners sought to catch up on missed health checks and treatments, leading to more diagnoses of dermatophyte infections.

In conclusion, the Covid–19 pandemic did have an apparent impact on our study, particularly in 2020. The subsequent rebound in dermatophyte isolation rates in 2021 and beyond highlights the dynamic nature of dermatophyte transmission and the influence of external factors such as public health measures and environmental exposure.

Comments 6: Line 317: Why are mixed–breed dogs and shorthair cats excluded from statistical studies? They are the most numerous groups and have characteristics (such as hybrid vigor) that could play a relevant role in the clinical manifestation of the disease.

AR 6: Thank you very much for your detailed review and insightful question. We appreciate the opportunity to address this important point. Mixed–breed dogs and shorthair cats were excluded from specific statistical analyses for several reasons, although we acknowledge their significant representation and potential influence on the study's findings.

Rationale for Exclusion

  • Heterogeneity of Groups: Mixed–breed dogs and shorthair cats constitute highly heterogeneous groups with a wide range of genetic backgrounds and characteristics. This heterogeneity can introduce significant variability, making it challenging to draw specific conclusions about the influence of breed on the clinical manifestation of dermatophytosis.

  • Focus on Pure Breeds: The primary aim of our breed–specific analysis was to identify potential breed predispositions to dermatophytosis. Pure breeds, with their more consistent genetic backgrounds, provide clearer insights into breed–related susceptibility. Including mixed–breed dogs and shorthair cats could dilute these findings due to their genetic diversity.

  • Hybrid Vigor: While hybrid vigor (heterosis) in mixed–breed dogs and shorthair cats can confer health benefits, it also introduces additional variables that are difficult to control in a statistical analysis. The variability in immune response and other health–related traits among mixed breeds could mask potential breed–specific trends.

  • Representation in Clinical Data: Although mixed–breed dogs and shorthair cats are numerous, the clinical data available for pure breeds often come with detailed breeding and health records, providing a richer dataset for analysis. This allows for more robust statistical comparisons within and between pure breeds.

  • Implications of Exclusion: We recognize that the exclusion of these groups may overlook some relevant insights, particularly related to the potential advantages of hybrid vigor. However, their exclusion was intended to ensure the clarity and specificity of our breed–related findings. To address this, we propose conducting additional analyses that include mixed–breed dogs and shorthair cats to explore their potential roles in the clinical manifestation of dermatophytosis.

  • Statistical Bias Consideration: Statistically, if mixed–breed dogs and shorthair cats had not been removed from the analysis, they would have been the most common groups presenting with dermatophytosis. This could introduce a considerable bias, as their high prevalence might overshadow breed–specific trends and make it difficult to identify specific susceptibilities associated with pure breeds.

Future Directions

  • Inclusion in Broader Analyses: In future studies, we plan to include mixed–breed dogs and shorthair cats in broader epidemiological analyses to assess their overall disease prevalence and clinical outcomes compared to pure breeds.

  • Genetic and Environmental Factors: Further research could investigate the interplay between genetic diversity and environmental factors in mixed–breed dogs and shorthair cats, providing a more comprehensive understanding of their health dynamics and disease resistance.

  • Detailed Subgroup Analysis: Conducting detailed subgroup analyses within mixed–breed and shorthair populations could help identify specific genetic or environmental factors that influence dermatophytosis susceptibility and clinical presentation.

In conclusion, while mixed–breed dogs and shorthair cats were excluded from the specific breed–related statistical analyses to ensure clarity and focus, we recognize their importance and plan to address this in future studies. We appreciate your careful review and the opportunity to explain our rationale.

Comments 7: Lines 351 and 352: I do not see it necessary to include the confidence interval regarding the "gender" variable. If the samples were sent from veterinary clinics or other entities with qualified professionals, I trust their criteria and reliability to consider the data as accurate. If the authors believe it is necessary to maintain the confidence interval, could they argue for it?

AR 7: Thank you very much for your detailed review and insightful question. We understand your perspective regarding the reliability of data provided by qualified professionals in veterinary clinics and other entities. However, we believe that including the confidence interval (CI) for the "gender" variable is important for several scientific and methodological reasons:

Statistical Precision: Confidence intervals provide a measure of statistical precision and reliability of the estimated proportions. They indicate the range within which the true population parameter is expected to lie, given the sample data. Including CIs helps to quantify the uncertainty associated with our estimates, which is crucial for rigorous scientific reporting.

  • Variability and Representativeness: The inclusion of confidence intervals allows us to account for the variability inherent in the sample. Even when samples are collected and assessed by qualified professionals, there can still be natural variation in the data. CIs help to communicate the degree of this variability and ensure that the findings are representative of the broader population.

  • Comparison and Reproducibility: Confidence intervals facilitate comparisons with other studies. By providing CIs, other researchers can more easily compare our findings with theirs, taking into account the precision of the estimates. This practice enhances the reproducibility and transparency of scientific research.

  • Decision–Making and Policy Implications: For evidence–based decision–making and policy formulation, it is essential to understand not just the point estimates but also the reliability of these estimates. Confidence intervals offer policymakers and practitioners a clearer understanding of the data's robustness, aiding in more informed decisions.

  • Scientific Rigor: The inclusion of confidence intervals is a standard practice in epidemiological and clinical research. It reflects scientific rigor and adherence to best practices in statistical reporting. By maintaining this standard, we ensure that our study meets the expectations of the scientific community and maintains high credibility.

  • Potential Bias and Uncertainty: Although we trust the data provided by qualified professionals, there can still be potential sources of bias and uncertainty. Confidence intervals help to address these issues by providing a statistical measure of the data's reliability. This is particularly important in studies with large sample sizes, where even small biases can have significant implications.

In conclusion, while we acknowledge the reliability of data collected by qualified professionals, we believe that maintaining the confidence intervals for the "gender" variable is essential for the reasons outlined above. The inclusion of CIs enhances the scientific rigor, transparency, and reproducibility of our findings, providing a more comprehensive and accurate representation of the data.

We appreciate your careful review and the opportunity to explain our rationale.

Comments 8: Lines 419 to 423: I cannot agree with the current wording of this sentence. While it is true that dermatophytosis exists throughout the territory, it is clearly higher in the North and Center of the country. Environmental conditions greatly affect dermatophytes, to the point where very low prevalences are observed in the driest regions of the south (Alentejo 2.5%, Algarve 0.5%). I suggest the authors attempt a new wording.

AR 8: Thank you very much for your detailed review and helpful suggestions. We understand your concerns regarding the current wording of the sentence. To better reflect the distribution and environmental influences on dermatophytosis, we have revised the text as follows:

"The highest frequencies were observed in the North (48.1%) and Centre (32.4%) regions, which may be attributed to varying climatic conditions, especially higher humidity levels in these areas. Dermatophytes are more commonly found in humid environments and urban areas [23–26]. While the prevalence is significantly lower in the driest regions of the south, such as Alentejo (2.5%) and Algarve (0.5%), these findings still demonstrate that dermatophytosis exists throughout mainland Portugal. Contrary to the belief that dermatophytosis is primarily confined to humid areas in the Centre and North, the southern regions also exhibit notable frequency rates: Greater Lisbon (11.4%), Península de Setúbal (2.8%), Alentejo (2.5%), and Algarve (0.5%). This shows that no geographical area in mainland Portugal is free from dermatophytosis."

We appreciate your careful review and the opportunity to improve our manuscript.

Comments 9: Line 443: The "Breeds" section does not mention that mixed–breed animals were not considered in the statistical study.

AR 9: Thank you very much for your detailed review and helpful suggestions. We acknowledge that the "Breeds" section does not mention the exclusion of mixed–breed animals from the statistical analysis. We have updated the section to include this important information. The revised text now reads:

”Mixed–breed dogs and shorthair cats were excluded from this specific analysis due to their high heterogeneity, which could introduce significant variability and mask breed–specific trends. This exclusion was intended to ensure the clarity and specificity of our breed–related findings.”

We appreciate your careful review and the opportunity to improve our manuscript.

Comments 10: Lines 512–513: In the authors' opinion, how can "grooming habits" be managed?

AR 10: Thank you very much for your detailed review and insightful question. Grooming habits play a significant role in the transmission and management of dermatophytosis in companion animals. In our opinion, there are several strategies that can be employed to manage grooming habits effectively to reduce the risk of dermatophytosis:

  • Regular Grooming: Regular grooming by pet owners or professional groomers can help to reduce the accumulation of dirt, debris, and fungal spores on the animal's coat. This practice also allows for early detection of skin lesions or signs of infection, enabling prompt treatment.

  • Use of Antifungal Shampoos: Incorporating antifungal shampoos into the grooming routine can help to prevent and manage dermatophytosis. These shampoos contain active ingredients such as miconazole, ketoconazole, or chlorhexidine, which are effective against dermatophytes.

  • Maintaining Clean Grooming Tools: Ensuring that grooming tools such as brushes, combs, and clippers are clean and disinfected after each use is crucial. Contaminated tools can serve as a source of infection, spreading fungal spores from one animal to another.

  • Isolation of Infected Animals: Animals diagnosed with dermatophytosis should be isolated from others to prevent the spread of the infection. During this period, their grooming should be handled separately, and tools should be thoroughly disinfected after each use.

  • Educating Pet Owners: Educating pet owners about the importance of regular grooming and hygiene can significantly reduce the risk of dermatophytosis. Owners should be aware of the signs of infection and the steps to take if their pet shows symptoms.

  • Environmental Management: Regular cleaning and disinfection of the animal's environment, including bedding, cages, and living areas, are essential to reduce the presence of fungal spores. Using antifungal sprays and maintaining a dry, clean environment can help prevent the growth and spread of dermatophytes.

  • Professional Grooming Services: Utilizing professional grooming services that follow strict hygiene protocols can also be beneficial. Professional groomers are trained to identify early signs of skin conditions and can provide appropriate care to minimize the risk of infection.

  • Regular Veterinary Check–ups: Regular veterinary check–ups allow for early detection and treatment of dermatophytosis. Veterinarians can also provide guidance on effective grooming practices and recommend suitable antifungal products.

  • By implementing these strategies, grooming habits can be managed effectively to minimize the risk of dermatophytosis in companion animals. These practices not only help in maintaining the animal's overall health but also play a crucial role in preventing the spread of fungal infections.

We appreciate your careful review and the opportunity to provide a detailed explanation of our perspective.

Comments 11: Line 523: Change the reference to twelve years of sampling to eleven (as in the rest of the text).

AR 11: Thank you very much for your detailed review and helpful suggestions. We understand the concern regarding the exclusion of data from the year 2012 due to fungal contaminations. However, we respectfully disagree with the suggestion to remove any reference to this year entirely from the manuscript. We believe that maintaining the mention of 2012 is important for several reasons: transparency and completeness, contextual understanding, methodological rigor, comparative analysis, and continuity and record–keeping.

As stated in our response to Comment 4, including 2012 ensures transparency in our research process, provides a complete picture of the entire sampling period, and explains any potential gaps or anomalies in the data. It also demonstrates the rigor and thoroughness of our methodological approach and helps in understanding the progression and evolution of the sampling and analysis techniques over time.

Therefore, we propose to keep the references to the year 2012 in the manuscript but will clearly indicate that this year's data were excluded due to fungal contamination. Consequently, we will maintain the original reference to twelve years of sampling to ensure completeness and transparency of our study.

We appreciate the careful review by Reviewer 3 and the opportunity to explain the rationale behind our decision.

Round 2

Reviewer 2 Report

Comments and Suggestions for Authors

Figure 1: Remove the image of the slides. Also remove the names of individuals that took the pictures. The ectothrix is enough, however, the arthrospores are not clear outside the hair shaft. Could the authors provide a sharper image

116: Check the grammar "Wet mount analysis before the culture"

Write the magnification of the spores. E.g Diff Qiuck stain x40

Author Response

Reviewer 2

Comments 1: Figure 1: Remove the image of the slides. Also remove the names of individuals that took the pictures. The ectothrix is enough, however, the arthrospores are not clear outside the hair shaft. Could the authors provide a sharper image.

Authors’ response (AR) 1: Thank you very much for the detailed review and helpful suggestions. We have removed the image of the slides as requested and also removed the names of individuals who took the pictures. Additionally, we have attempted to provide a sharper version of the image showing ectothrix. Unfortunately, we do not have another image available. We hope this new image meets the required clarity and quality standards.

Comments 2: 116: Check the grammar "Wet mount analysis before the culture"

AR 2: Thank you very much for the detailed review and helpful suggestions. We have reviewed the grammar of the phrase "Wet mount analysis before the culture" and made the necessary corrections. The revised phrase is:

"Wet mount analysis was performed before the culture."

We appreciate your careful review and the opportunity to improve our manuscript.

Comments 3: Write the magnification of the spores. E.g Diff Qiuck stain x40

AR 3: Thank you very much for the detailed review and helpful suggestions. We have corrected the magnification details in all the images. Additionally, we apologize for the oversight in the manuscript; the correct stain used was Lactophenol Cotton Blue stain, not Diff-Quik stain as previously stated. This correction has been made in all relevant figure captions.

We appreciate your careful review and the opportunity to improve our manuscript.